# SRBD1 facilitates chromosome segregation by promoting topoisomerase IIα localization to mitotic chromosomes

Courtney A. Lovejoy [1] ✉, Sarah R. Wessel[1,5], Rahul Bhowmick [1], Yuki Hatoyama [2,3], Masato T. Kanemaki [2,3,4], Runxiang Zhao[1] & David Cortez [1] ✉

Accurate sister chromatid segregation requires remodeling chromosome architecture, decatenation, and attachment to the mitotic spindle. Some of these events are initiated during S-phase, but they accelerate and conclude during mitosis. Here we describe SRBD1 as a histone and nucleic acid binding protein that prevents DNA damage in interphase cells, localizes to nascent DNA during replication and the chromosome scaffold in mitosis, and is required for chromosome segregation. SRBD1 inactivation causes micronuclei, chromatin bridges, and cell death. Inactivating SRBD1 immediately prior to mitotic entry causes anaphase failure, with a reduction in topoisomerase IIα localization to mitotic chromosomes and defects in properly condensing and decatenating chromosomes. In contrast, SRBD1 is not required to complete cell division after chromosomes are condensed. Strikingly, depleting condensin II reduces the severity of the anaphase defects in SRBD1-deficient cells by restoring topoisomerase IIα localization. Thus, SRBD1 is an essential genome maintenance protein required for mitotic chromosome organization and segregation.

Maintaining genome integrity during cell division requires both complete duplication of the genome and segregation of sister chromatids into daughter cells. These processes depend on the coordinated action of hundreds of proteins and are remarkably accurate. However, when errors occur, they often cause disease, including cancer.

Genome architecture plays crucial roles in both DNA replication and chromosome segregation. An initial level of organization comes from the packaging of DNA into chromatin using histone and non-histone DNA binding proteins. This packaging is temporarily perturbed during DNA replication[1,2]. DNA unwinding generates topological changes, including supercoiling, and a requirement for topoisomerase enzymes to relieve the torsional strain[3–5]. Type II topoisomerases catalyze the removal of additional topological barriers, such as DNA knots and catenanes, to maintain replication fork movement and complete DNA synthesis[6].

Interphase chromatin is further organized by a process of loop extrusion into topologically associated domains (TADs) that can regulate gene expression, replication timing, and DNA repair[7,8]. Topoisomerase IIβ localizes at the base of these loops and may resolve topological problems that arise from loop extrusion, but also makes these sites prone to double strand breaks and chromosome rearrangements[9]. Longer-distance folding between TADs, driven again by local changes in topology, compacts and organizes chromosomes into territories that influence inter- and intra-chromosomal interactions with important implications for genome stability[10–12].

[1]Department of Biochemistry, Vanderbilt University School of Medicine, Nashville, TN, USA. [2]Department of Chromosome Science, National Institute of Genetics, Research Organization of Information and Systems (ROIS), Yata 1111, Mishima, Shizuoka, Japan. [3]Graduate School for Advanced Studies, SOKENDAI, Yata 1111, Mishima, Shizuoka, Japan. [4]Department of Biological Science, Graduate School of Science, The University of Tokyo, Tokyo, Japan. [5]Present address: BPGbio, Framingham, MA, USA. ✉e-mail: courtney.lovejoy@vanderbilt.edu; david.cortez@vanderbilt.edu

Loop extrusion is also important in chromosome condensation and segregation during mitosis. Mitotic chromosomes are formed as a linear array of chromatin loops radially organized around a central scaffold of non-histone proteins[13–15]. A small set of proteins comprise this chromosome scaffold, and show a distinctive localization pattern along the central axes of mitotic chromosomes[16–19]. Condensin complexes and topoisomerase IIα (topo IIα) are major components of this protein scaffold[20]. Condensin II generates large loops compacting the chromosomes lengthwise, from which condensin I generates nested loops to compact chromosomes laterally[21–23]. Condensin-mediated loop extrusion recruits and directs the activity of topo IIα to drive chromosome individualization and induce further axial shortening[24–30]. These changes in chromosome architecture are essential to withstand mitotic spindle forces and segregate chromatids at anaphase[31–34]. Inactivating condensin or topo IIα causes defects in mitotic chromosome organization, resulting in the persistence of DNA intertwines and anaphase chromatin bridges that cause genome damage and instability[21,35,36].

Here we identify S1 RNA Binding Domain Containing Protein 1 (SRBD1) as an essential genome maintenance protein required for chromosome segregation. Previous studies linked *SRBD1* polymorphisms to an increased glaucoma risk[37,38], and an SRBD1-ALK fusion was described in adenocarcinoma patients[39,40], but little functional information exists for this largely unstudied protein. We show that SRBD1 is recruited to nascent chromatin in S-phase and localizes to the central axes of chromosomes in mitosis. Silencing SRBD1 in cycling cells causes DNA damage and cell death. Inactivating SRBD1 in late G2 phase cells causes anaphase failure due to a marked reduction in topo IIα

chromatin association and the persistence of massive chromosome entanglements. The severity of the anaphase defects can be partly suppressed by inactivating condensin II, which restores the localization of topo IIα to mitotic chromosomes. Furthermore, SRBD1 inactivation in prometaphase, after chromosomes have condensed, yields a largely normal anaphase. These results indicate that SRBD1 has a critical role in establishing the proper chromosome architecture in early mitosis for successful cell division.

## Results

### SRBD1 is an uncharacterized histone and nucleic acid binding protein that is enriched on nascent DNA during replication

We previously utilized iPOND (isolation of proteins on nascent DNA) combined with quantitative mass spectrometry to identify proteins enriched at replication forks and on nascent chromatin in diverse cell types[41]. One notable protein in this catalog is SRBD1, a largely uncharacterized protein with putative nucleic acid binding motifs. Its relative abundance on nascent DNA compared to mature chromatin is similar to that observed for MCM2, a component of the replicative helicase (Fig. 1a). Although SRBD1 is enriched on nascent DNA, the protein remains associated with chromatin throughout the cell cycle (Fig. 1b).

The predicted AlphaFold structure for SRBD1 closely resembles that of the *P. aeruginosa* toxin expression (Tex) and *S. cerevisiae* suppressor of Ty6 (Spt6) proteins (Supplementary Fig. 1a). Tex has four putative nucleic acid binding domains and may regulate transcription[42]. Spt6 is a histone chaperone and transcription elongation factor that also functions in DNA replication[43–46]. Spt6 is important

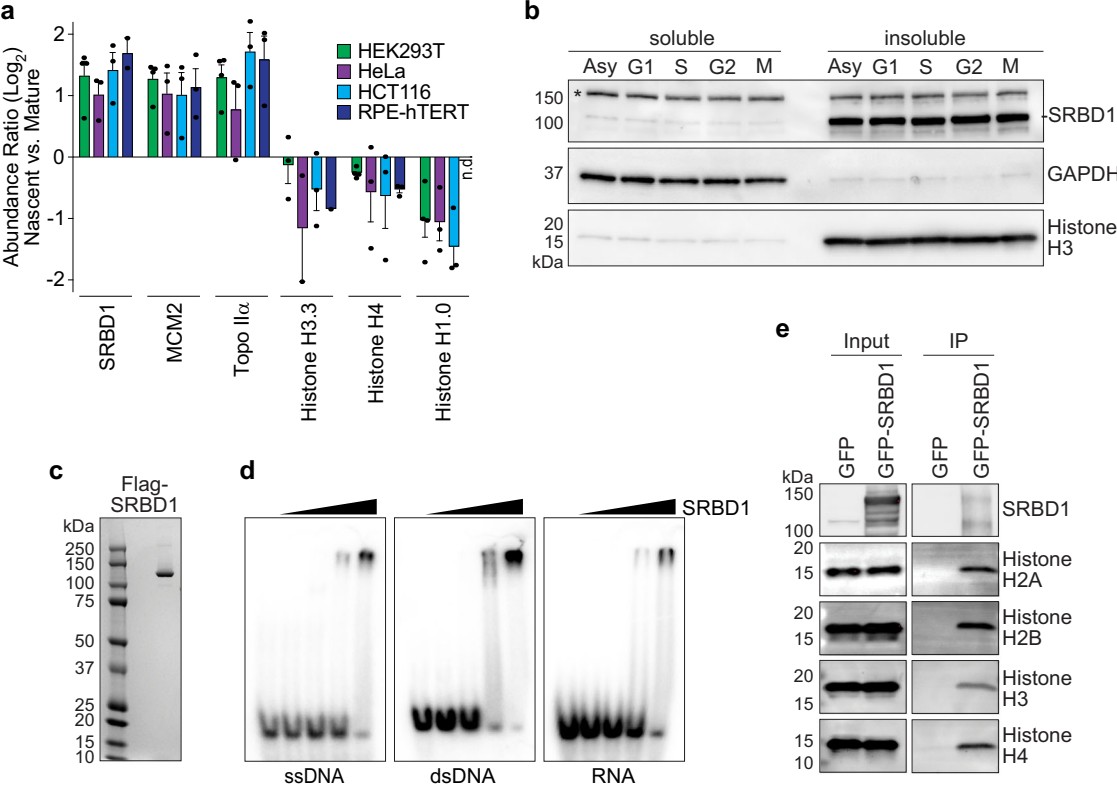

**Fig. 1 | SRBD1 is a histone and nucleic acid binding protein. a** iPOND proteomics data[41] reveals that SRBD1 is enriched on nascent DNA similar to MCM2 and topo IIα during DNA replication, in contrast to histones which are depleted (n.d., not detected). The graph displays the mean and SEM, with each data point representing a biological replicate. **b** Immunoblot for SRBD1 on subcellular fractionations from asynchronously (Asy) growing U2OS, and cells synchronized in G1, S, G2, and mitosis (M). GAPDH and Histone H3 immunoblots identify the soluble and chromatin-associated fractions, respectively. An asterisk denotes a non-specific band. **c** Coomassie-stained gel showing purification of Flag-SRBD1 from HEK293T cells. **d** Electrophoretic mobility shift assay with the indicated nucleic acids and 0–200 nM of Flag-SRBD1. **e** Immunoblots of histones co-immunoprecipitated with GFP or GFP-SRBD1 from nuclease-treated HEK293T nuclear extracts. Data are representative of at least two independent experiments (**b**–**e**). Source data are provided as a Source Data file.

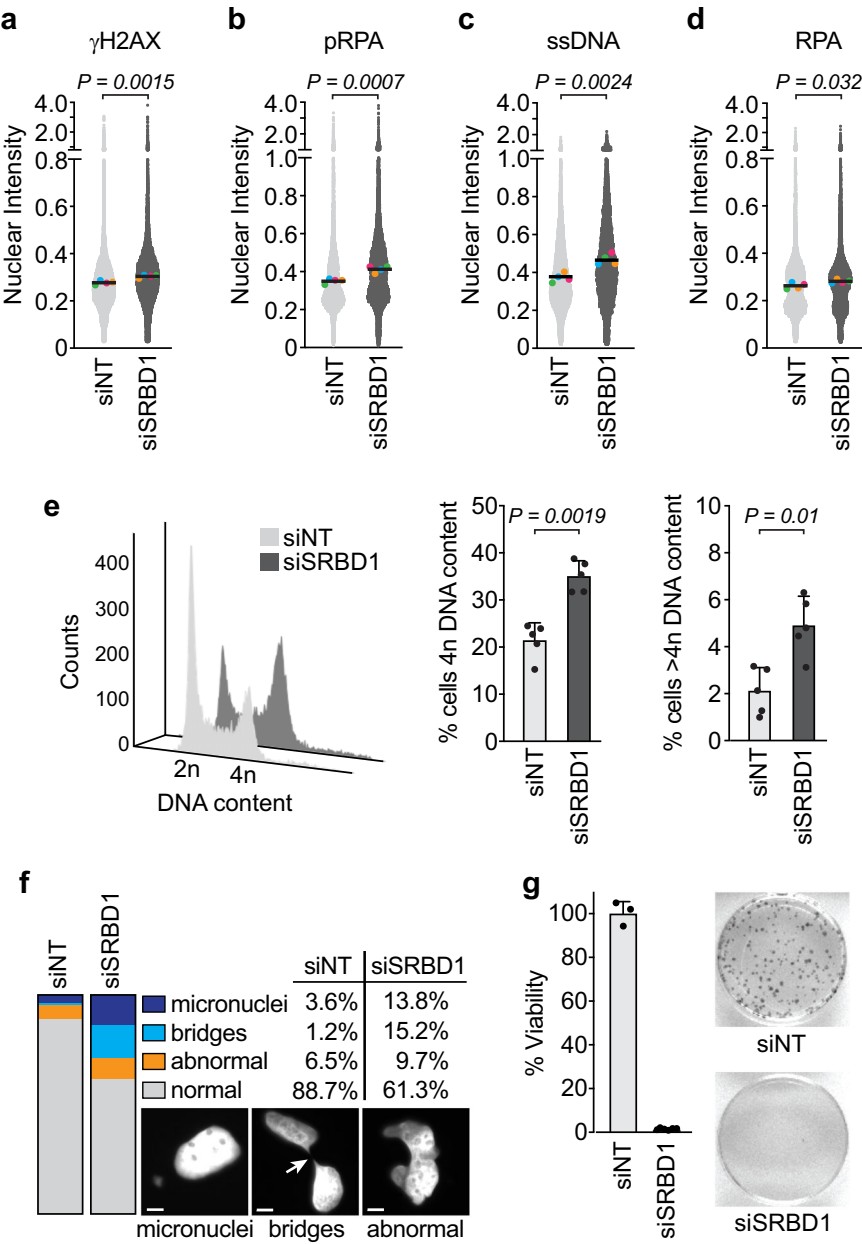

**Fig. 2 | Inactivation of SRBD1 causes genome instability. a–d** Levels of γH2AX, pS4,S8-RPA, ssDNA, and insoluble RPA were measured in U2OS cells by immunofluorescence imaging 48 h after transfection with non-targeting (NT) or SRBD1 siRNAs. Each gray data point represents the nuclear intensity in one cell (arbitrary units x 10⁷; total cells analyzed ≥ 11,947). The colored data points represent the mean nuclear intensity from each biological replicate (*n* = 4). Black bars are the mean of the replicate experiments, and significance was determined using a two-sided, unpaired t-test comparing the means of the replicate experiments. **e** Cell cycle distributions were analyzed by flow cytometry 48 h after transfection of U2OS cells with NT and SRBD1 siRNAs. A representative cell cycle profile (left; 10,000 gated cells) and quantitation of the fraction of cells with 4n or greater than 4n DNA content (right) is shown. The graphs display the mean +/− SD of biological replicates (*n* = 5) and significance was determined using a two-sided, paired t-test. **f** U2OS nuclear morphology was visualized by DAPI staining 48 h after transfection with NT and SRBD1 siRNAs across two independent experiments (total cells analyzed ≥ 145). Scale bars represent 5 μm. **g** The viability of U2OS cells following siRNA depletion of SRBD1 was measured by colony forming ability. Quantitation (left) and a representative colony image (right) are shown. The graph displays the mean +/− SD of biological replicates (*n* = 3 for siNT; *n* = 6 for siSRBD1). Source data are provided as a Source Data file.

for preventing DNA damage and ensuring genome stability, possibly by regulating chromatin structure[47–49]. SRBD1 lacks the region of Spt6 required for interactions with RNA polymerase II, and RNA sequencing revealed few significant changes in gene expression after RNAi depletion of SRBD1 (Supplementary Fig. 1b, c), suggesting SRBD1 does not regulate transcription. However, we found that SRBD1 can bind single stranded (ss) and double stranded (ds) DNA, as well as RNA (Fig. 1c, d). Additionally, SRBD1 co-immunoprecipitates with histones H2A, H2B, H3, and H4 from nuclease-treated cell extracts (Fig. 1e).

## SRBD1 has genome maintenance activities

To evaluate a potential genome maintenance function of SRBD1, we examined loss of function phenotypes following siRNA-mediated depletion in U2OS cells (Supplementary Fig. 1d). SRBD1-deficient cells displayed elevated γH2AX and RPA phosphorylation, markers of DNA damage and replication stress (Fig. 2a, b). This DNA damage signaling was accompanied by increases in ssDNA, measured with native BrdU staining and elevated levels of insoluble RPA (Fig. 2c, d). SRBD1 depletion also caused an accumulation of cells with 4n and greater

than 4n DNA content (Fig. 2e). Collectively, this suggests SRBD1 has an important role in preventing DNA damage and genome instability.

When examining the SRBD1-depleted cells, we found that there was a substantial increase in micronuclei and chromatin bridges in interphase cells (Fig. 2f). We also observed an increase in cells with oddly shaped nuclei and potential binucleation, consistent with the appearance of cells with greater than 4n DNA content by flow cytometry. The SRBD1-silenced cells exhibited a striking reduction in cell growth and were unable to form colonies, in agreement with the common essential designation for SRBD1 in the DepMap database (Fig. 2g and Supplementary Fig. 1e). These outcomes are characteristic of cells experiencing chromosome segregation errors during mitosis, which could arise from replication problems that persist into mitosis or from defective mitotic processes.

### Degradation of SRBD1 results in extensive anaphase chromatin bridges

To identify when during the cell division cycle SRBD1 function is critical and what the direct and immediate consequences of SRBD1 loss are, we turned to a degron system that facilitates rapid SRBD1 inactivation. We generated an auxin-inducible SRBD1 degron using the AID2 system in HCT116 cells expressing an OsTIR1(F74G) mutant[50]. The mAID and mClover tags were inserted at the 5' end of both SRBD1 alleles using CRISPR-Cas9-directed gene editing. The expression of mAID-mClover-SRBD1 (mAC-SRBD1) was confirmed by immunoblotting, and in two separate clones (designated c1 and c2) the tagged protein shows the expected change in molecular weight and is rapidly degraded following addition of the synthetic auxin analog, 5-phenyl-indole-3-acetic acid (5-Ph-IAA; Fig. 3a and Supplementary Fig. 2a). As expected, degradation of SRBD1 completely prevents colony formation, while the viability of the parental HCT116 cells is unaffected by 5-Ph-IAA treatment (Supplementary Fig. 2b).

Inactivation of SRBD1 in the HCT116 cells caused a modest increase in γH2AX by 4 h after 5-Ph-IAA addition in one clonal cell line, and within one hour in the second clone, with more significant increases by 16 and 24 h (Fig. 3b and Supplementary Fig. 2c, d). Flow cytometry analysis revealed that SRBD1 degradation increased the proportion of cells with 4n and greater than 4n DNA content, while rapidly inducing apoptosis. This increase in 4n DNA content was accompanied by only a modest increase in histone H3 S10 phosphorylation (a marker of mitotic cells) and was partially relieved by inhibition of cell cycle checkpoint kinases, indicating that a subset of the 4n cells were arrested at the DNA damage checkpoint in G2 (Fig. 3c, d and Supplementary Fig. 2e).

Based on the iPOND data localizing SRBD1 to nascent DNA during DNA replication, and the DNA damage accumulation and G2 checkpoint arrest after SRBD1 inactivation, we hypothesized that the loss of function phenotypes could be tied to incomplete or defective replication triggering chromatin bridges and nuclear abnormalities. If so, then depletion of SRBD1 in G2 phase cells should not have any effect until the following S-phase. To test this hypothesis, we degraded SRBD1 in G2 synchronized cells, released into mitosis and followed them over time. Surprisingly, we immediately observed evidence of mitotic defects including anaphase chromatin bridges, observable with DAPI staining, and ultrafine DNA bridges (UFBs), which lack histones and are only visible by staining for associated proteins such as PICH (Fig. 3e–g)[51,52]. Approximately 35–40% of anaphases from SRBD1-deficient cells displayed UFBs while 60% of anaphases in both degron clones showed dramatic bridging of chromatinized DNA, often with multiple thick chromatin bridges and sometimes no distinct bridge but rather a mass of chromatin between the segregating chromatids (Fig. 3e, g). Anaphase bridges can arise more frequently from specific loci, including centromeres and ribosomal DNA. However, immunostaining for Centromere protein B (CENPB) and Upstream Binding Factor (UBF) to mark these

loci indicate most chromatin bridges do not arise from these genomic regions, consistent with a more widespread problem in sister chromatid entanglements (Supplementary Fig. 3a–d). The extensive anaphase bridging observed after G2 phase degradation of SRBD1 suggests a direct role for SRBD1 in facilitating chromosome segregation during mitosis, independent of its function in S-phase.

### SRBD1 functions in early mitosis to prevent anaphase failure

To gain further insight into the mitotic problems arising from degradation of SRBD1, we stably expressed H2B-mCherry in the parental and two degron clones to facilitate live cell imaging of the chromosomes. Each cell line was synchronized in late G2 before treating with DMSO or 5-Ph-IAA for one hour, then released into mitosis and imaged by live cell spinning-disk confocal microscopy. Anaphase chromatin bridges were again observed in both clones after inactivating SRBD1. Even more prominent was a complete anaphase failure. Chromosomes condensed, aligned at the metaphase plate, and anaphase was initiated as the chromatids were visibly (and very briefly) pulled toward opposite poles, but massive entanglements appeared to prevent sister chromatid segregation (Fig. 4a, b, Supplementary Fig. 4, Supplementary Movies 1–4). The attempt at sister chromatid segregation was quickly aborted and the segregating DNA masses collapsed together, started decondensing, and ultimately produced abnormal nuclei reminiscent of that observed following siRNA depletion of SRBD1 (Fig. 4a–i and Supplementary Fig. 4–i). To verify these phenotypes were not dependent on the CDK1 inhibition used to achieve G2 synchronization, asynchronously growing cells were treated with 5 Ph-IAA for one hour and anaphase defects were characterized by live imaging only for cells that entered mitosis within the next hour. The timing of these experiments ensures that only cells in which SRBD1 was degraded in late G2 are analyzed. The predominant phenotypes again are anaphase failure and chromatin bridges (Fig. 4c). A small fraction of cells in one degron clone also showed an apparent metaphase arrest (defined as >100 min in metaphase without an attempt at anaphase; Fig. 4a-iv, c).

To determine when in mitosis SRBD1 function is critical, we synchronized cells in prometaphase by treatment with nocodazole, induced SRBD1 degradation with the addition of 5 Ph-IAA, then released cells and analyzed anaphase progression. Degradation of SRBD1 during prometaphase nearly eliminated the anaphase failure phenotype in both degron clones. The subset of anaphases that displayed chromatin bridges also had a less severe phenotype, with fewer and/or less bulky chromatin bridges relative to those observed after SRBD1 degradation in G2 (Fig. 5a, b and Supplementary Fig. 5a). Notably, most anaphases were normal when SRBD1 was degraded at prometaphase, something rarely observed with SRBD1 degradation in G2 phase cells (Figs. 4b, 5b). This suggests that SRBD1 has critical functions in mitosis prior to prometaphase.

The prometaphase delay in nocodazole (a total of 2 h) could also have provided additional time for the chromatin entanglements to be resolved. To test this idea, we repeated the SRBD1 degradation in G2 phase cells but included a prometaphase delay in nocodazole for 2 h before releasing cells and analyzing anaphase progression. Additional time in prometaphase also rescued the most severe anaphase failure phenotypes resulting from G2 phase degradation of SRBD1, but these cells retained more chromatin bridges than what was observed with SRBD1 degradation during prometaphase (Fig. 5c, d and Supplementary Fig. 5b). Thus, while delaying mitotic progression can aid in the resolution of chromosome entanglements, degradation of SRBD1 after chromosome condensation largely prevents segregation errors, indicating that SRBD1 has an essential function early in mitosis.

### The mitotic spindle in SRBD1-deficient cells appears normal until anaphase initiation

Two processes are initiated in early mitosis—spindle formation and chromosome condensation. We assessed spindle integrity on fixed

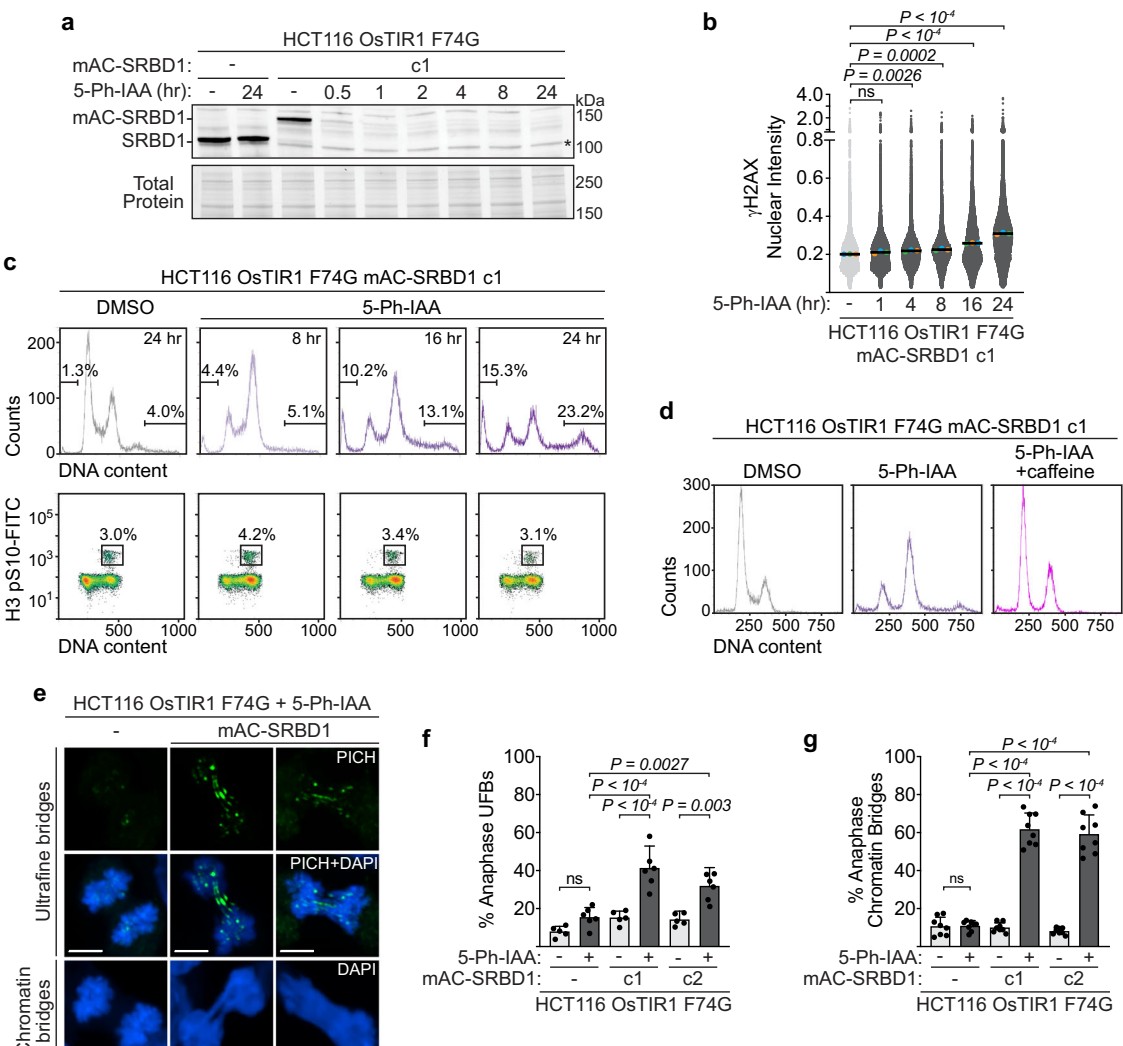

**Fig. 3 | Acute degradation of SRBD1 results in anaphase bridges. a** Immunoblot analysis of SRBD1 (parental clone) and mAID-mClover-SRBD1 (mAC-SRBD1, clone 1) in asynchronous cells treated with DMSO (24 h) or 1 µM 5-Ph-IAA for the indicated amounts of time. Bio-Rad Stain-Free total protein was used as a loading control. A cross-reacting protein that migrates just below the untagged SRBD1 protein is denoted by an asterisk. **b** Asynchronously growing cells were treated with DMSO (24 h) or 1 µM 5-Ph-IAA for the indicated amounts of time and γH2AX was measured by immunofluorescence imaging. Each gray data point represents the nuclear intensity in one cell (arbitrary units x $10^7$; total cells analyzed ≥19,388). The colored data points represent the mean of each biological replicate ($n = 3$) and black bars represent the mean of the three replicates. Significance was determined using a one-way ANOVA with Dunnett's multiple comparisons test comparing the means of replicate experiments. **c** Asynchronously growing cells were treated with DMSO (24 h) or 1 µM 5 Ph-IAA for the indicated amounts of time

and cell cycle distributions (from 25,000 gated cells) were analyzed by flow cytometry. pH3 staining was used to differentiate mitotic and G2 phase cells. **d** Asynchronously growing cells were treated with DMSO or 1 µM 5-Ph-IAA -/+ 8 mM caffeine for 16 h and cell cycle distributions were analyzed by flow cytometry on 25,000 gated cells. **e–g** G2 phase synchronized cells were treated with DMSO or 1 µM 5 Ph-IAA for 1 h, released into mitosis, and anaphase bridges were assessed by immunostaining. **e** Representative images of ultrafine anaphase bridges (detectable by PICH immunostaining, green), and anaphase chromatin bridges (identified with DAPI staining, blue). Scale bars represent 5 µm. The percentage of anaphases (total cells analyzed ≥360) with PICH or BLM-coated ultrafine bridges (**f**) and chromatin bridges (**g**) was quantified. Graphs display the mean +/− SD of biological replicates ($n = 5$ or 6 for UFBs and $n = 8$ for chromatin bridges). Significance was determined using a one-way ANOVA with Sidak's multiple comparisons test. Source data are provided as a Source Data file.

mitotic cells stained for tubulin after G2 phase degradation of SRBD1. Monopolar spindles were not observed in either degron clone, and multipolar spindles were rarely observed in both control and SRBD1-deficient cells. There was no incidence of unfocused or splayed spindle poles, and spindle length measurements from metaphase cells revealed no change following SRBD1 degradation (Supplementary Fig. 6a, b). We also expressed tubulin-GFP in cells with H2B-mCherry to visualize spindle dynamics throughout mitosis. Asynchronously growing cells were treated with 5-Ph-IAA for one hour and only cells that entered mitosis within the next hour were evaluated to ensure the analysis was restricted to cells in which SRBD1 was inactivated in late G2. Spindle formation appeared to progress normally in both control and SRBD1-

deficient cells until the beginning of anaphase. Microtubules nucleated from two centrosomes as the chromosomes began condensing, the centrosomes moved to opposite poles, and a mitotic spindle formed (Supplementary Fig. 6c, d and Supplementary Movies 5, 6). The chromosomes aligned along the spindle equator and a pulling force to separate the sister chromatids was observed, suggesting successful kinetochore-microtubule attachments and the initiation of anaphase (panels marked by an asterisk in Supplementary Fig. 6c, d). The ability of SRBD1-deficient cells to arrest in prometaphase when treated with nocodazole also indicates the mitotic checkpoint is intact and SRBD1-deficient cells can sense proper attachments. We conclude that the spindle forms and functions properly through the end of metaphase.

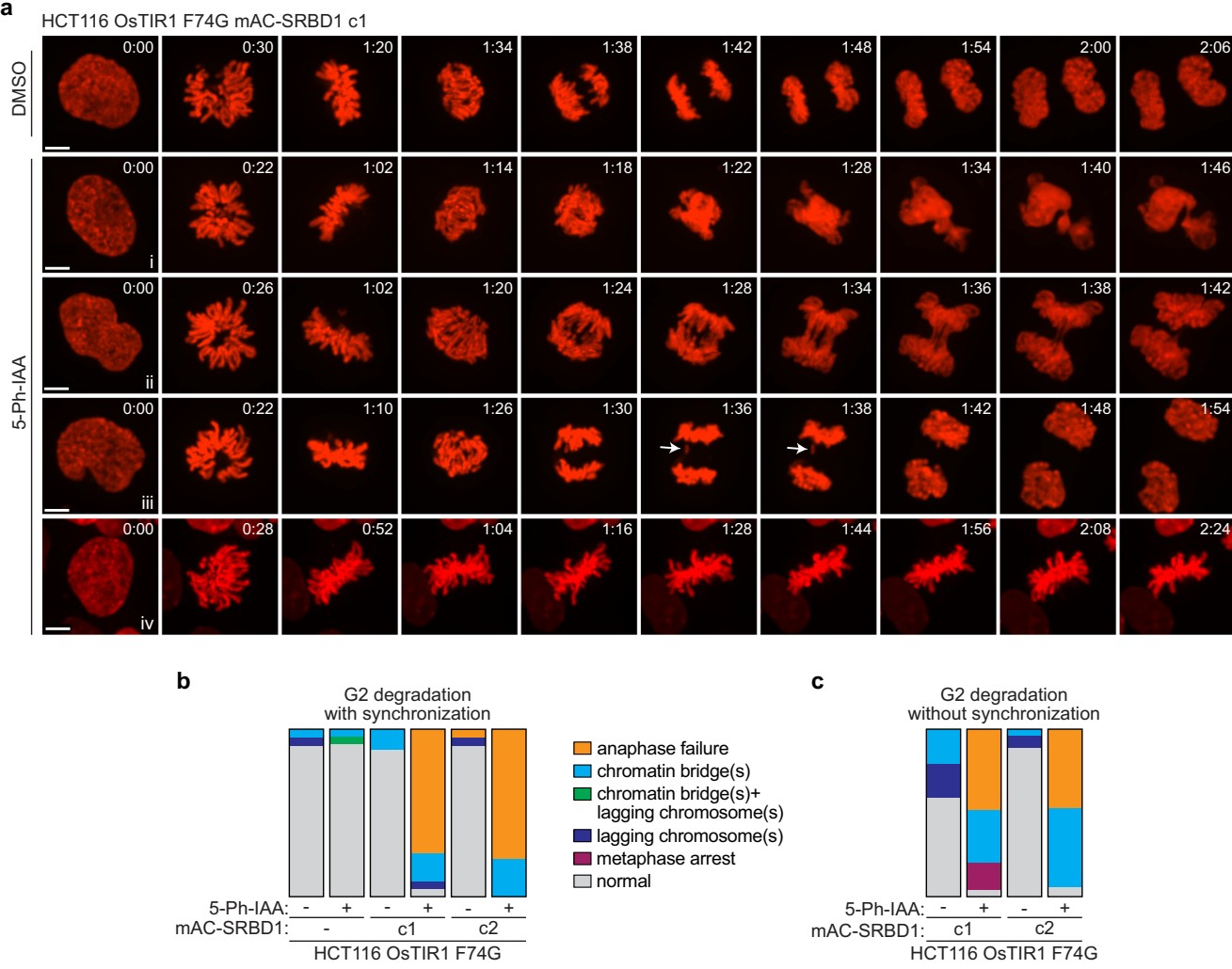

**Fig. 4 | SRBD1 is required for chromosome segregation. a** Still pictures derived from live imaging of SRBD1 degron cells expressing histone H2B-mCherry. A comparable time point in prophase is shown for each, with subsequent panels displaying prometaphase, metaphase, the initiation of anaphase, and comparable time points thereafter. A normal anaphase in DMSO-treated cells is shown, along with representative images of anaphase failure (i), chromatin bridges (ii), lagging chromosomes (iii; denoted by an arrow), and metaphase arrest (iv; defined as >100 min in metaphase without an attempt at anaphase) observed after 5-Ph-IAA-induced degradation of SRBD1. Scale bars represent 5 μm. **b** G2 phase synchronized cells were treated with DMSO or 1 μM 5 Ph-IAA for 1 h, released into mitosis, and anaphase defects derived from live-cell imaging were quantified (16–23 cells analyzed). **c** Asynchronously growing cells were treated with DMSO or 1 μM 5 Ph-IAA for 1 h and only cells entering mitosis within the next hour were analyzed (to ensure experiment was restricted to SRBD1 degradation in late G2 phase cells). Anaphase defects derived from live-cell imaging were quantified (19–29 cells analyzed). Bars represent the fraction of cells displaying the indicated phenotypes (**b**, **c**). Source data are provided as a Source Data file.

During anaphase in control cells, the chromosomes were first pulled towards the poles, the poles separated as the spindle elongated, and the separated chromosomes then decondensed (Supplementary Fig. 6c and Supplementary Movie 5). In contrast, the spindle poles are pulled together in most SRBD1-deficient cells as the chromatin masses collapsed and decondensed (Supplementary Fig. 6d and Supplementary Movie 6). In the subset of SRBD1-deficient cells that display less severe phenotypes with discernable anaphase chromatin bridges, the spindle is capable of elongating and pulling chromosomes toward opposite poles (Supplementary Fig. 6e).

### SRBD1 influences mitotic chromosome architecture through regulation of topo IIα localization

The failure to resolve sister chromatid entanglements or to sufficiently compact chromosomes can lead to extensive chromatin bridges at anaphase. These processes are mediated by a specific subset of non-histone proteins distributed axially along metaphase chromosomes referred to as the mitotic chromosome scaffold. Importantly, live cell imaging showed that SRBD1 localized along the length of condensing chromosomes in early mitosis (Supplementary Fig. 7a). Furthermore, examination of SRBD1 localization on fixed metaphase spreads revealed a striking axial localization pattern that is absent when SRBD1 is degraded (Fig. 6a and Supplementary Fig. 7b). Metaphase spreads from SRBD1-deficient cells collected after 2 h of demecolcine treatment also showed unusually elongated chromosomes, suggesting a defect in axial chromosome compaction (Fig. 6b, c). No overt chromatid entanglements are observed in this context, consistent with the ability of an extended prometaphase to rescue most chromosome segregation defects after SRBD1 degradation (Fig. 5c).

The axial localization of SRBD1 on metaphase chromosomes and defects in chromosome compaction after SRBD1 degradation prompted us to examine condensin and topo IIα, two major components of the protein scaffold that regulates mitotic chromosome architecture. Notably, when examining the iPOND data, we noticed that one of the most highly correlated proteins with SRBD1 across all previously reported iPOND-SILAC-mass spectrometry datasets is topo

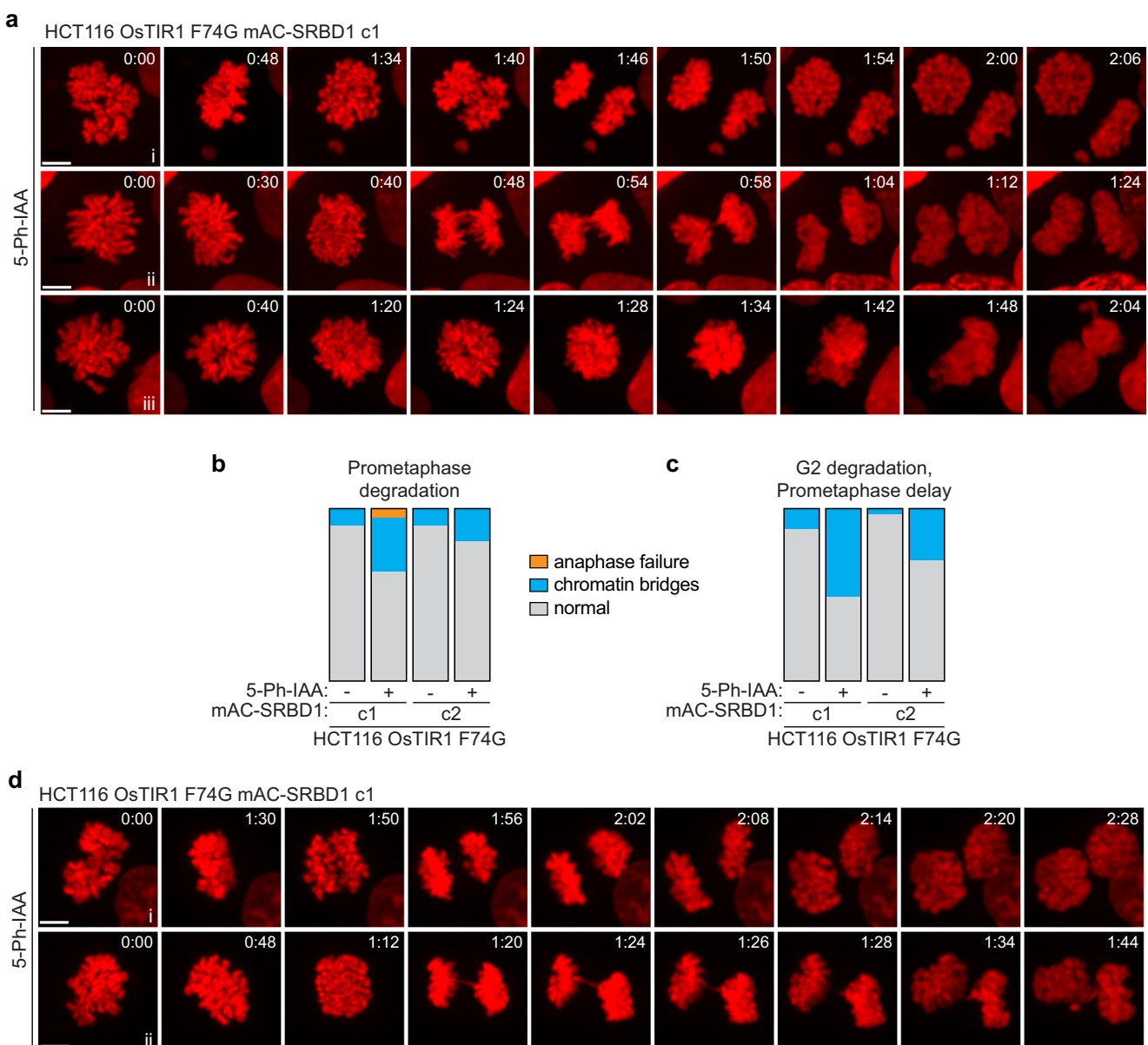

**Fig. 5 | SRBD1 function is critical in early mitosis to prevent anaphase failure.**
**a** Still pictures derived from live imaging of histone H2B-mCherry after SRBD1 degradation during prometaphase. The first panel displays the prometaphase-synchronized cells at the start of imaging, with subsequent panels showing metaphase, the initiation of anaphase, and comparable time points after initiation. Normal anaphase progression (i), chromatin bridges (ii), and rarely anaphase failure (iii) were observed after 5 Ph-IAA-induced degradation of SRBD1 during prometaphase. **b** G2 phase synchronized cells were released into mitosis in the presence of 100 ng/ml nocodazole for 2 h to enrich for prometaphase cells. Degradation of SRBD1 was induced during the second hour of nocodazole treatment with the addition of 1 µM 5 Ph-IAA (or DMSO for controls), cells were released from the nocodazole block and anaphase defects derived from live-cell imaging were quantified (19–29 cells analyzed). **c** G2 phase synchronized cells were treated with DMSO or 1 µM 5 Ph-IAA for 1 h and released into nocodazole for 2 h. Cells were then released from the nocodazole block and anaphase defects derived from live-cell imaging were quantified (25–31 cells analyzed). Bars represent the fraction of anaphases displaying the indicated phenotypes (**b**, **c**). Anaphases with lagging chromosomes were equally abundant in all samples due to the nocodazole treatment and were scored as normal anaphase progression. **d** Still pictures derived from live imaging of histone H2B-mCherry after degradation of SRBD1 in G2 phase synchronized cells followed by a 2 h delay in nocodazole. The first panel displays the prometaphase-synchronized cells at the start of imaging, with subsequent panels showing metaphase, the initiation of anaphase, and comparable time points after. Normal anaphase progression (i) and chromatin bridges (ii) were observed after 5-Ph-IAA-induced degradation of SRBD1. All scale bars represent 5 µm. Source data are provided as a Source Data file.

IIα, along with CDYL, RBBP4, and CBX2, three components of complexes that promote repressive chromatin environments (Supplementary Fig. 7c). This correlation in abundance on nascent DNA across multiple experimental conditions suggests there may be a physical or functional relationship between SRBD1 and topo IIα[41,53]. Since we were unable to observe co-immunoprecipitation between SRBD1 and topo IIα in the absence of DNA, we examined whether SRBD1 loss impacted topo IIα localization. Immunostaining for the condensin subunit SMC2

showed the expected accumulation along the central axes of metaphase chromosomes in the presence and absence of SRBD1 (Fig. 6d and Supplementary Fig. 7d). Topo IIα also localized to the central axes of metaphase chromosomes and accumulated at centromeres in the parental control cells (Fig. 6d). However, the mitotic localization of topo IIα was dramatically altered by SRBD1 degradation. Topo IIα was largely absent from chromosome arms and displayed weaker centromere accumulation in both SRBD1-deficient clones (Fig. 6d, e). This

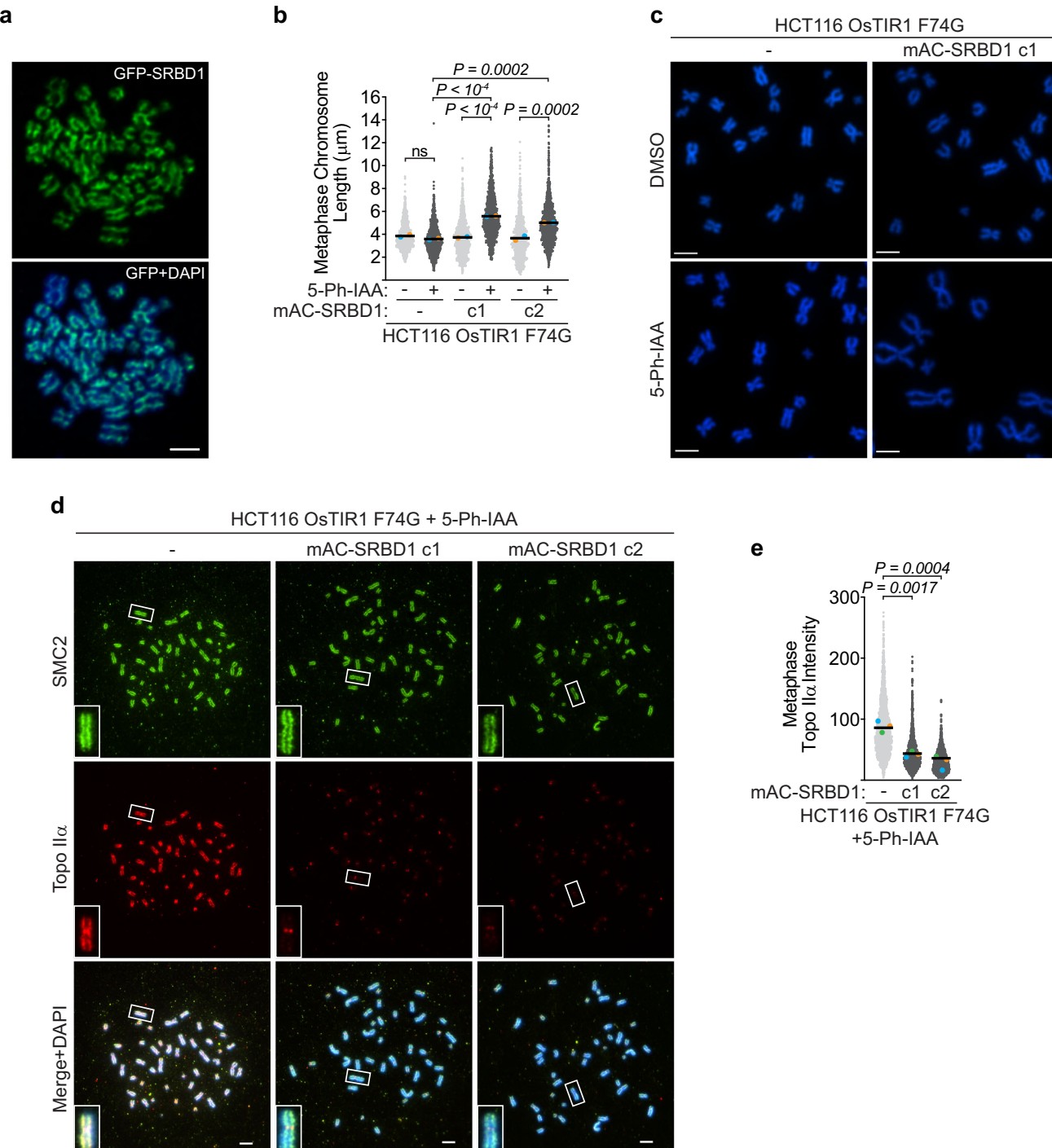

**Fig. 6 | SRBD1 localizes to the central axes of mitotic chromosomes and is required for the mitotic localization of topo IIα. a** Representative immunostaining of SRBD1 (GFP, green) on metaphase chromosomes, observed in at least two independent experiments. DAPI is in blue. Scale bar represents 5 μm.
**b** Chromosome length measurements from DAPI-stained metaphase spreads collected after G2 phase degradation of SRBD1 (DMSO or 1 μM 5-Ph-IAA for 1 h) and 2 h in demecolcine. Each gray data point represents the length of one metaphase chromosome (total chromosomes analyzed ≥772). The colored data points represent the mean chromosome length from each biological replicate (*n* = 2). Black bars represent the mean of the two replicates and significance was determined using a one-way ANOVA with Sidak's multiple comparisons test of the means.
**c** Representative images of DAPI-stained metaphase spreads from cells treated with DMSO or 5 Ph-IAA. Scale bars represent 5 μm. **d**, **e** G2 phase synchronized cells were

treated with 1 μM 5-Ph-IAA for 1 h and released into demecolcine for 35–45 min. The localization of mitotic scaffold proteins on metaphase chromosome spreads was examined by immunostaining. **d** Representative images of SMC2 and topo IIα immunostaining on metaphase chromosome spreads. DNA is stained with DAPI. Scale bars represent 5 μm. **e** Quantitation of topo IIα intensity on metaphase chromosome spreads using CellProfiler. Each gray data point represents the topo IIα intensity on one metaphase chromosome (arbitrary units; total chromosomes analyzed ≥ 1944). The colored data points represent the mean chromosomal intensity from each biological replicate (*n* = 3). Black bars represent the mean of the three replicates and significance was determined using a one-way ANOVA with Dunnett's multiple comparisons test comparing the means of the replicate experiments. Source data are provided as a Source Data file.

change in localization was not attributable to variations in the DAPI intensity of metaphase chromosomes (Supplementary Fig. 7e), nor was the topo IIα protein targeted for degradation simply by being in proximity of SRBD1 (Supplementary Fig. 7f).

We next asked whether SRBD1 functions in the same pathway for controlling mitotic chromosome structure as either of the condensin complexes, which are known to direct the localization and activity of topo IIα[25,27,30,31]. Cells depleted of the condensin I subunit CAP-H continued to predominantly undergo anaphase failure after degradation of SRBD1 (Fig. 7a, b). Surprisingly, condensin II depletion dramatically improved the mitotic defects associated with SRBD1 inactivation and even yielded some anaphases with no apparent defects (Fig. 7a, siCAP-D3; Supplementary Movies 7–11). Knockdown of either condensin II subunits CAP-H2 or CAP-D3 significantly reduced the fraction of cells that failed anaphase (Fig. 7a, b and Supplementary Fig. 8a). This reduction in anaphase failure produced a corresponding increase in the less severe anaphase chromatin bridges, with the cells largely able to segregate sister chromatids (Fig. 7c). Importantly, this rescue was not due to an indirect effect on cell proliferation or mitotic progression. Flow cytometry analysis revealed almost no change in the cell cycle profiles of CAP-H2 or CAP-D3-depleted cells at this time point, except for a small increase in the percentage of mitotic cells marked by histone H3 S10 phosphorylation (Supplementary Fig. 8b). A minor delay in mitosis is not unexpected after condensin depletion but is unlikely to explain the large improvement in mitotic outcomes in SRBD1-deficient cells since a similar increase in mitotic cells was observed after CAP-H depletion, which did not improve anaphase progression. Furthermore, the extent of the increase in mitotic cells did not correlate with the extent of mitotic improvement, since depletion of CAP-H2 produced a greater increase in mitotic cells while CAP-D3 depletion produced better mitotic outcomes. Although condensin II depletion improved the mitotic defects observed in the first mitosis after SRBD1 degradation, it did not rescue the viability defect resulting from inactivation of SRBD1 (Supplementary Fig. 8c, d).

To understand why condensin II depletion improved the SRBD1-deficient phenotypes, we examined topo IIα localization. Surprisingly, despite condensin II being important for topo IIα localization, the combined inactivation of condensin II and SRBD1 significantly improved topo IIα localization to mitotic chromosomes, indicating a synthetic rescue (Fig. 7d, e and Supplementary Fig. 8e, f). Although the total intensity of topo IIα on metaphase chromosomes after the combined loss of condensin II and SRBD1 approached that of control cells, the localization along chromosome arms still appeared weaker and more irregular. The improvement in topo IIα localization when condensin II is inactivated in SRBD1-deficient cells likely explains the improvement in chromosome disentanglement and segregation in these cells, although some defects remain.

Given the localization of SRBD1 along the central axes of mitotic chromosomes, the loss of insoluble topo IIα from mitotic chromosomes in SRBD1-deficient cells, the apparent defects in chromosome individualization and compaction following SRBD1 inactivation, and the genetic interaction with condensin II, we conclude that SRBD1 is a component of the mitotic chromosome scaffold that is essential for properly shaping and segregating sister chromatids.

## Discussion

Accurate chromosome segregation requires a dramatic reorganization of chromatin that resolves sister chromatid entanglements, compacts chromosomes, and makes them capable of withstanding mitotic spindle forces. These changes in chromosome architecture are driven by condensin complexes and depend on the localization and activity of topo IIα, which collectively form a protein scaffold along the center of mitotic chromosomes. Here, we identify SRBD1 as a component of the mitotic scaffold that is essential for chromosome segregation. SRBD1 is a DNA and histone binding protein that localizes to the central axes of mitotic chromosomes as well as nascent chromatin during replication. Both are genomic locations and times during cell division where DNA is undergoing significant changes in topology and packaging. SRBD1 prevents DNA damage accumulation in interphase and acts during early mitosis to shape mitotic chromosomes and promote the proper localization of topo IIα. Live cell imaging revealed extensive chromosome segregation problems when SRBD1 is inactivated in late G2 phase, with sister chromatids attempting to pull apart but failing to segregate and quickly collapsing together and decondensing. The abundance of chromatin entanglements that do not appear to originate from a particular locus suggests the persistence of sister chromatid catenanes, and the abnormally long metaphase chromosomes indicate defects in chromosome compaction as well. SRBD1 function is critical in prophase, when dramatic changes are occurring in mitotic chromosome structure that require topo IIα catalytic activity. Topo IIα accumulation is markedly reduced at centromeres and nearly absent from chromosome arms in mitotic cells after SRBD1 inactivation, likely explaining the defects in chromosome compaction and severe entanglements that block chromosome segregation.

The anaphase failure in SRBD1-deficient cells can be partially rescued by inactivation of condensin II, likely due to the restoration of mitotic topo IIα localization. Condensin and topo IIα are two of the few proteins needed to assemble mitotic chromosomes[54]. They work cooperatively to compact sister chromatids, drive decatenation along the chromosome arms, and individualize the chromosomes to make them competent for segregation in anaphase[24,25,28,31,55,56]. How inactivation of both condensin II and SRBD1 restores topo IIα localization to mitotic chromosomes, when loss of either alone dramatically impairs topo IIα localization, is unclear. However, this concept of synthetic rescue is not uncommon, and it suggests a complex relationship between these proteins that ultimately determines whether the sister chromatids are properly disentangled prior to segregation. The inactivation of both condensin II and SRBD1 may allow an alternative pathway to productively shape mitotic chromosomes. Another possibility is that the combined loss of condensin II and SRBD1 may correct an imbalance created by loss of either alone. Since degradation of SRBD1 during prometaphase did not produce severe anaphase defects and condensin II depletion (but not condensin I) reduced the anaphase failure, SRBD1 may be particularly important in the early steps of chromatid individualization and compaction. Delaying anaphase in SRBD1-deficient cells also resolved the most severe anaphase defects, suggesting that the reduced topo IIα on chromosomes in SRBD1-deficient cells can eventually decatenate the sister chromatids. Nevertheless, the delay still left elongated mitotic chromosomes and suggests a defective chromosome architecture remains in the SRBD1-deficient cells.

Given the ability of SRBD1 to bind DNA and histones, it is noteworthy that chromatin remodeling proteins are also required for the in vitro assembly of mitotic chromatids[54], and the proper organization of DNA and histones is necessary for the productive actions of condensin complexes and topo IIα[57–60]. An interesting possibility is that SRBD1 could influence large-scale changes in genome organization through local changes in nucleosomes and DNA topology. The DNA binding and loop extrusion activities of condensin are stimulated by positive supercoiling, which merges neighboring coils of DNA into a single supercoiled loop that favors the recruitment of additional condensin proteins[61]. The positive supercoiling stabilized by condensin-mediated loop extrusion also creates topological stress that recruits topo IIα and influences the directionality of topo IIα-catalyzed DNA movements to favor decatenation[24,62]. Interactions between SRBD1 and nucleic acids and histones may promote changes in chromatin structure and DNA topology in a way that is needed to bring

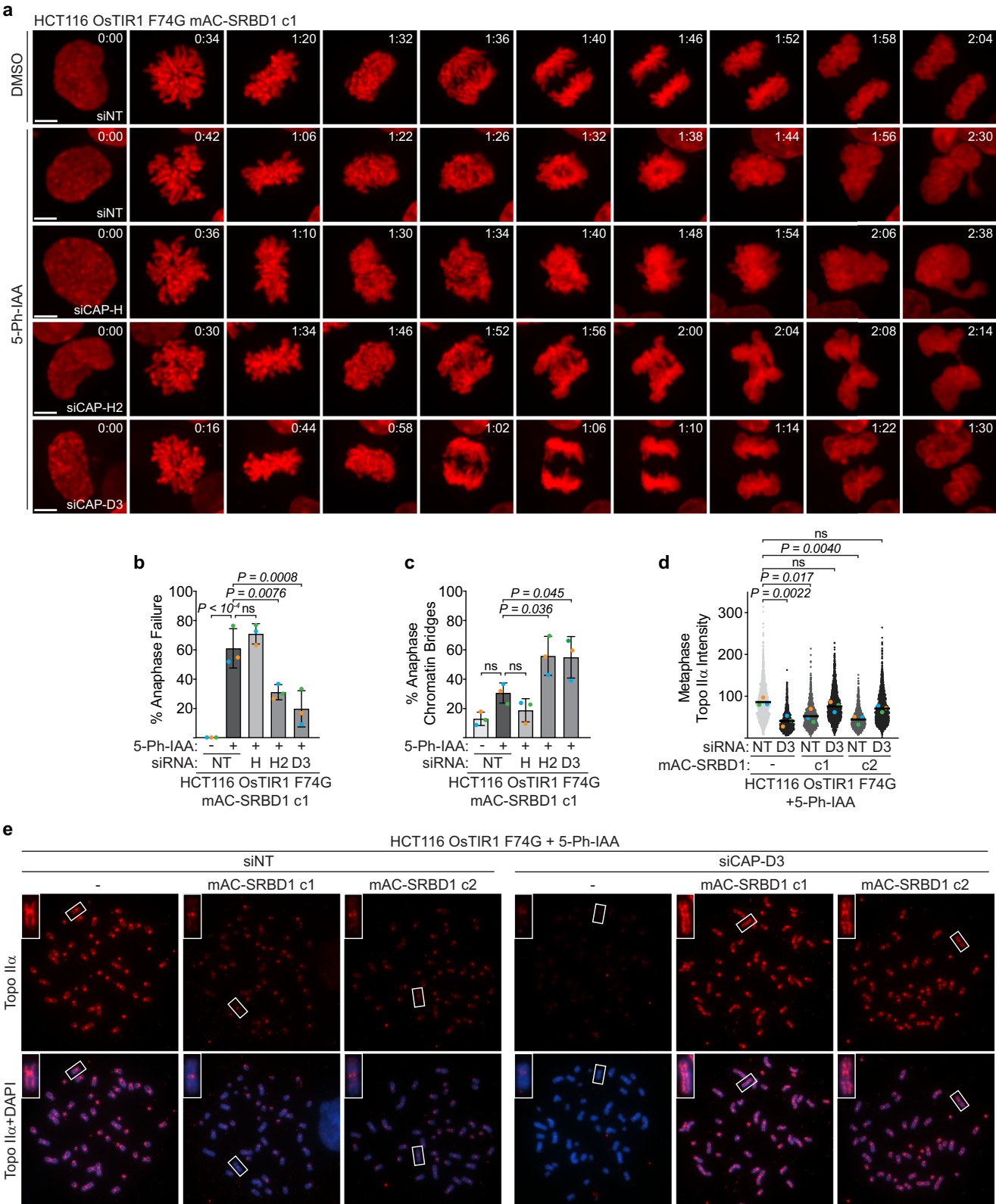

topo IIα to mitotic chromosomes and complete the disentanglement of the sister chromatids. It is also possible that SRBD1 has functions at specific genomic locations, although the amount of bridging, staining pattern, and lack of locus specificity suggests a more global function.

While our data suggest the severe chromosome segregation problems in SRBD1-deficient cells arise primarily from defects in shaping and disentangling chromosomes in early mitosis, we cannot exclude the possibility of an additional function in anaphase to promote the proper action of the mitotic spindle. The assembly and function of the mitotic spindle appear normal in SRBD1-deficient cells through metaphase. There are no obvious spindle shape abnormalities, we observe normal chromosome congression, and the mitotic checkpoint activity that monitors spindle attachments appears functional. Although we often observe a clear attempt to pull the sister

**Fig. 7 | Condensin II depletion reduces mitotic failure in SRBD1-deficient cells by restoring topo IIα localization. a–c** Cells were transfected with siRNAs targeting condensin I (CAP-H) or condensin II (CAP-H2 and CAP-D3) and anaphase defects were examined by live cell imaging. Non-targeting (NT) siRNA was used as a control. **a** Still pictures derived from live imaging of histone H2B-mCherry after transfection with the indicated siRNAs. A comparable time point in prophase is shown for each, with subsequent panels displaying prometaphase, metaphase, the initiation of anaphase, and comparable time points thereafter. **b, c** Quantitation of anaphase failure (**b**) and anaphase chromatin bridges (**c**) following treatment with 1 μM 5 Ph-IAA (or DMSO) for 1 h in G2 phase synchronized cells (blue/orange data points) or in asynchronously growing cells (green data points). For the asynchronous samples, cells entering mitosis within the next hour after the 5 Ph-IAA treatment were analyzed to ensure only cells in which SRBD1 degradation was restricted to G2 phase cells were scored. Graphs display the mean +/− SD for biological replicates ($n = 3$; total cells analyzed ≥ 74) and significance was determined using a one-way ANOVA with Dunnett's multiple comparisons test. **d, e** Cells transfected with NT or CAP-D3 siRNAs were synchronized in G2 phase for SRBD1 degradation (1 μM 5 Ph-IAA for 1 h) and released into demecolcine for 35–45 min to assess the mitotic localization of topo IIα. **d** Quantitation of topo IIα intensity on metaphase chromosome spreads. Each gray/black data point represents the topo IIα intensity on one metaphase chromosome (arbitrary units; total chromosomes analyzed ≥ 1271). The colored data points represent the mean chromosomal intensity for each biological replicate ($n = 3$). Black bars represent the mean of the three replicates and significance was determined using a one-way ANOVA with Dunnett's multiple comparisons test comparing the means of the replicate experiments. **e** Representative images of topo IIα immunostaining on metaphase chromosome spreads. DNA is stained with DAPI. All scale bars represent 5 μm. Source data are provided as a Source Data file.

chromatids toward opposite poles at the initiation of anaphase, the spindle does not elongate in SRBD1-deficient cells that display the anaphase failure phenotype. Instead, the spindle poles are pulled together at a point when spindle elongation and increased separation between the poles is critical for chromosome segregation. Therefore, defects in spindle function could contribute to the anaphase failure in SRBD1-deficient cells. However, we favor the interpretation that the primary cause of the chromosome segregation errors is a failure in decatenation for the following reasons. First, a subset of SRBD1-deficient cells can largely segregate their chromosomes with normal spindle movements, but they retain extensive chromatin bridges. Second, there is a visible attempt at chromosome segregation in many SRBD1-deficient cells that undergo anaphase failure, with a pulling force that briefly produces a modest degree of chromosome separation. Third, the mislocalization of topo IIα in SRBD1-deficient cells is consistent with the persistence of unresolved DNA catenanes. Fourth, the reduction in anaphase failure by condensin II depletion in SRBD1-deficient cells is accompanied by a rescue in topo IIα localization to mitotic chromosomes, suggesting improved decatenation facilitates anaphase progression. Fifth, delaying cells in prometaphase with nocodazole largely rescues the anaphase failure in SRBD1-deficient cells. Additional time might readily allow a reduced amount of topo IIα on chromosomes to resolve the entanglements, but it is unclear how additional time in prometaphase would rescue an anaphase spindle defect. Sixth, SRBD1 localizes to the central axes of mitotic chromosomes and binds DNA and histones. We have not observed SRBD1 localization to the spindle. Thus, multiple lines of evidence suggest that SRBD1 is required for topo IIα-mediated chromosome decatenation. In the absence of SRBD1, the persistence of massive DNA entanglements likely blocks chromosome segregation and consequently prevents appropriate spindle movements.

The chromosome condensation and segregation defects caused by late G2 phase inactivation of SRBD1 highlight a function for SRBD1 in early mitosis to productively shape the chromosomes. In addition, SRBD1 has important functions in interphase cells. It localizes to chromatin throughout the cell cycle and is enriched on nascent DNA in replicating cells, where chromatin is being re-assembled and topoisomerases are acting to relieve torsional stress. Inactivating SRBD1 in interphase cells generates hallmarks of DNA damage and replication stress, which ultimately yield a cell cycle checkpoint-dependent arrest in G2 phase. Importantly, loop extrusion is a common mechanism regulating genome architecture in interphase as well as mitosis, again involving topoisomerases, condensin II, and other closely related proteins[11,63,64]. Condensin II and topoisomerase activities on newly replicated DNA are also important for resolving entanglements and initiating structural changes critical for chromosome segregation in mitosis[65,66]. Improper reassembly of chromatin after DNA synthesis in SRBD1-deficient cells could readily produce unwound or damaged DNA, consistent with the elevation in RPA/ssDNA and γH2AX observed after SRBD1 inactivation. Thus, SRBD1 may regulate chromosome architecture throughout the cell cycle. Further studies will be needed to understand the mechanisms by which SRBD1 acts in both S-phase and mitosis. Nonetheless, the striking phenotypes caused by inactivation of this essential and largely uncharacterized protein suggest there is still much to learn about the intricacies of these two fundamental processes.

## Methods

### Cell culture

U2OS (ATCC, HTB-96; osteosarcoma, human female origin) and HEK293T cells (ATCC, CRL-3216; epithelial, human female origin) were grown in Dulbecco's modified Eagle's medium (DMEM, Invitrogen) supplemented with 7.5% fetal bovine serum (FBS; Atlanta Biologicals). HCT116 cells (ATCC, CCL-247; carcinoma, human male origin) were grown in McCoy's 5 A medium (Invitrogen) supplemented with 10% FBS and GlutaMax (Invitrogen). Details of the iPOND-SILAC mass spectrometry and data analysis have been published[41]. All cell lines were purchased from ATCC, cultured under standard conditions at 37 °C, 5% CO2, and regularly tested for mycoplasma (Universal Mycoplasma Detection kit; ATCC).

### RNA interference

U2OS cells were transiently transfected with 40 nM siRNA using DharmaFECT Duo (Horizon Discovery Biosciences) and HCT116 cells were transiently transfected with 20–25 nM siRNA using Lipofectamine RNAiMax (ThermoFisher Scientific) for 48 h, according to the manufacturer's instructions. siRNAs were obtained from Horizon Discovery Biosciences or Qiagen (SRBD1, Horizon J-018144-09, J-018144-10, J-018144-11, J-018144-12; CAP-H, Horizon L-012853-01; CAP-H2, Horizon L-016186-01; CAP-D3, Horizon L-016186-01; AllStars negative control, Qiagen 1027281) and target protein depletion was confirmed by immunoblotting.

### Endogenous tagging of SRBD1 with mAID-mClover

HCT116 mAID-mClover-SRBD1 cell lines were generated by CRISPR-Cas9-mediated genome editing, as previously described in refs. 50,67. Briefly, we designed a CRISPR-Cas9 plasmid for targeting the SRBD1 locus (5'-ACCTGTACTTTCGCTCTTCT(TGG)-3'). A donor plasmid harboring mAID-mClover and a selection marker (HygroR) with two homology arms (approximately 500 bp each) was constructed. HCT116 cells stably expressing OsTIR1(F74G) from the AAVS1 locus were transfected with the CRISPR and donor plasmids. Clones surviving hygromycin selection were screened for biallelic insertion of the tagging construct by genomic PCR, and expression of the fusion protein was confirmed by immunoblotting. Two distinct clonal cell lines are utilized throughout and designated as c1 and c2.

### Immunoblotting

Whole-cell lysates were prepared using Igepal lysis buffer (50 mM Tris-HCl, pH 7.4, 150 mM NaCl, 1% Igepal CA-630, 1 mM EDTA, pH 8.0)

supplemented with 1 mM dithiothreitol (DTT), 1 mM sodium fluoride, 1 mM sodium vanadate, 1 mM phenylmethylsulfonyl fluoride (PMSF), 1 μg/mL aprotinin, 1 μg/mL leupeptin, Pierce Universal Nuclease (ThermoFisher Scientific), and 1 mM MgCl₂. Equivalent amounts of protein (Bio-Rad DC protein assay) were separated by SDS-PAGE and immunoblotted. Bio-Rad Stain-Free total protein was used for loading controls. The following antibodies were used: SRBD1 (1:1000; Bethyl, A305-121A); GAPDH (1:5000; MilliporeSigma, MAB374); histone H2A (1:2000; abcam, ab18255); histone H2B (1:2000; abcam, ab1790); histone H3 (1:2000; abcam, ab10799); histone H4 (1:2000; abcam, ab31830); Topo IIα (1:2000; Santa Cruz Biotechnology, sc-166934); SMC2 (1:2000; abcam, ab10412); CAP-H (1:1000; Bethyl, A300-603A); CAP-H2 (1:1000; abcam, ab200659); CAP-D3 (1:1000; Bethyl, ab200659).

### Cell synchronization and subcellular fractionation

Asynchronously growing cells were synchronized in G1 with 2 mM thymidine (MilliporeSigma, 89270) for 16 h. Thymidine-synchronized cells were washed with warm media and released for 5 h to enrich for S phase cells or released 5 h and then incubated in 100 ng/ml nocodazole (MilliporeSigma, M1404) for 10 h to synchronize at mitosis. Cells were synchronized in G2 phase with 10 μM Ro-3306 (ApexBio, A8885) for 16 h. Cell synchronization was confirmed by flow cytometry. For the subcellular fractionation, cells harvested by trypsinization were resuspended in Buffer A (100 mM NaCl, 300 mM sucrose, 3 mM MgCl₂, 10 mM PIPES, pH 6.8, 1 mM EGTA, 0.2% Triton X-100, 1 mM sodium vanadate, 1 mM sodium fluoride, and protease inhibitor cocktail (Roche, 04693159001)) and incubated on ice for 5 min. The insoluble (chromatin-associated) material was pelleted at 1400 g for 4 min and the supernatant retained as the soluble protein fraction. Chromatin fractions were washed once with Buffer A, resuspended in Buffer B (50 mM Tris-HCl, pH 7.5, 150 mM NaCl, 5 mM EDTA, 1% Triton X-100, 0.1% SDS, 1 mM sodium vanadate, 1 mM sodium fluoride, protease inhibitor cocktail (Roche)) for 10 min on ice, and sonicated (Bioruptor, Diagenode) for 15 cycles of 30 sec on/ 30 sec off. Samples were centrifuged at max speed for 5 min and equivalent amounts of protein were separated by SDS-PAGE and immunoblotted.

### Protein purification

Flag-SRBD1 was purified from HEK293T cells. Cell pellets were resuspended in cytoplasmic lysis buffer (10 mM Tris-HCl, pH 7.9, 0.34 M sucrose, 3 mM calcium chloride, 2 mM magnesium acetate, 0.1 mM EDTA, 1 mM DTT and protease inhibitor cocktail (Roche)) containing 0.1% Igepal for 5 min on ice. After centrifugation, the nuclei were washed once with cytoplasmic lysis buffer and resuspended in nuclear lysis buffer (50 mM HEPES, pH 7.9, 10% glycerol, 150 mM potassium acetate, 1.5 mM magnesium chloride, 0.1% Igepal, 1 mM DTT and protease inhibitor cocktail). Nuclei were homogenized and treated with Pierce Universal Nuclease for 2 h at 4 °C. The lysate was diluted 3-fold with wash buffer (20 mM HEPES, pH 7.9, 150 mM potassium chloride, 0.5 mM EDTA, 0.1% Triton X-100, 10% glycerol, 1 mM DTT, and protease inhibitor cocktail, (Roche)), cleared by high-speed centrifugation, and filtered (Pall AP-4425). Magnetic Flag beads (MilliporeSigma) were washed 3 times with wash buffer and incubated with the cleared lysate for 3.5 h at 4 °C. The beads were washed 4 times with wash buffer and Flag-SRBD1 protein was eluted with 3X Flag peptide (MilliporeSigma) in 1X TBS containing 5% glycerol at 4 °C overnight.

### Gel shift assays

For the SRBD1 electrophoretic mobility shift assays, 5 nM ³²P-labelled substrate (39-40 nucleotide mixed sequence for all ss/ds DNA and RNA) was incubated with increasing amounts of Flag-SRBD1 (25–200 nM) in 10 or 20 μL reactions containing 20 mM Tris-HCl, pH 8, 150 mM NaCl, 2.5% glycerol and 1 mM DTT. The reactions were incubated at room temperature for 30 min and separated on a 10% 1X

TBE gel at 75 V for ~1 h. Gels were dried and imaged using a Typhoon. The sequences of the ssDNA and ssRNA substrates are CTGAG-GAAATGCGTGGCGGGTGATTGGCGGGCTGGATAAA and CGCAU-CAAGGGUUAUCACAGUCACGAUCCUAGUAGUGCU, respectively. An unlabeled, complementary DNA was annealed to the labeled, ssDNA oligonucleotide to generate the dsDNA substrate.

### Co-immunoprecipitation

HEK293T cells were transfected with GFP-SRBD1 or GFP-NLS using polyethylenimine (PEI Max, Polysciences) and nuclear extracts were prepared in the presence of Pierce Universal Nuclease (ThermoFisher Scientific). GFP-tagged constructs were immunoprecipitated using GFP-Trap magnetic agarose beads (Chromotek) and co-precipitating proteins were analyzed by SDS-PAGE and immunoblotting.

### Immunofluorescence imaging in interphase cells

Cells were plated in 96-well clear-bottom plates and analyzed 48 h after siRNA transfection or at the indicated time points after the addition of DMSO/5-Ph-IAA. Where insoluble protein is specified, cells were pre-extracted for 2–5 min on ice (20 mM HEPES, pH 7.0, 50 mM NaCl, 3 mM MgCl₂, 300 mM sucrose, and 0.5% Triton X-100), fixed in 3% paraformaldehyde, and washed with PBS. For antigens not requiring pre-extraction, cells were fixed and washed. All were permeabilized for 10 min on ice with the 0.5% Triton buffer detailed above and blocked for 1 h at room temperature (1 mg/mL BSA, 5% goat serum, 0.1% Triton X-100, and 1 mM EDTA in 1X PBS). Primary antibody incubations with γH2AX (1:1000; MilliporeSigma, 05-636), pS4,S8-RPA (1:200; Bethyl, A300-245A), and RPA32 (1:250; abcam, ab2175) were performed at 4 °C overnight in blocking buffer. After incubation with fluorescently labeled secondary antibodies, nuclei were stained with DAPI (Sigma, D9542). Plates were imaged on a Molecular Devices ImageXpress Micro and the nuclear integrated intensity was quantified using the MetaXpress Multi Wavelength Cell Scoring software application module. For native BrdU staining, cells were labeled with 10 μM BrdU for 24 h before the cells were pre-extracted and fixed, as described above. Blocking, antibody incubation (MoBU-1 1:200; ThermoFisher Scientific, B35128), and image acquisition was also performed as described above. The quantitation of micronuclei, cytoplasmic DNA bridges, and abnormal nuclei was manually scored on asynchronously growing, DAPI-stained cells in the experiments described above.

### Analysis of anaphase bridges and mitotic spindle

Cells plated on glass coverslips were synchronized in G2 phase by treatment with 6 μM Ro-3306 for 16 h (ApexBio, A8885). After a 1 h treatment with DMSO or 1 μM 5 Ph-IAA (Tocris Bioscience, 739210), cells were washed four times with pre-equilibrated media and released into mitosis. For the analysis of chromatin and ultrafine anaphase bridges, cells were pre-extracted (0.1% Triton X-100 in 20 mM PIPES, pH 6.8, 1 mM MgCl₂, 10 mM EGTA) and fixed (0.05% Triton X-100, 4% formaldehyde, 20 mM PIPES, pH 6.8, 1 mM MgCl₂, 10 mM EGTA) at 42–45 min after release, as described previously[68]. Samples were blocked in PBSAT (3% BSA, 0.5% Triton X-100 in 1x PBS) and incubated with PICH (1:40; Millipore, 04-1540), BLM (1:400; abcam, ab2179), CENPB (1:200; abcam, ab25734), or UBF (1:200; Santa Cruz Biotechnology, sc-13125) primary antibodies diluted in PBSAT at 4 °C overnight. For mitotic spindle staining, cells were fixed at 42–45 min post-release for 10 min in ice cold methanol. Samples were blocked in PBSAT and incubated with tubulin antibody (DM1A 1:200; Millipore Sigma, T9026) diluted in PBSAT at 4 °C overnight. Cells for both anaphase bridges and mitotic spindle analysis were washed three times with PBSAT (10 min each), incubated with fluorescently labeled secondary antibodies for 1 h at room temperature, washed again with PBSAT (three washes, 15 min each), and mounted using ProLong Gold containing DAPI (ThermoFisher Scientific, P36941). Images were

obtained using a 40X oil objective (Nikon Eclipse Ti). Anaphase bridges were scored manually, and metaphase spindle length was measured using Nikon Elements software.

## Immunofluorescence of metaphase chromosome spreads

Cells were synchronized in G2 phase by treatment with 6 µM Ro-3306 for 16 h (ApexBio, A8885), washed four times with pre-equilibrated media, and released into media containing 0.1 µg/ml demecolcine (MilliporeSigma, D1925) with DMSO or 1 µM 5 Ph-IAA (Tocris Bioscience, 739210) for 2 h. Mitotic cells were harvested by shake-off, pelleted by centrifugation, and swollen by incubation in pre-warmed 75 mM KCl for 20 min at 37 °C. For immunostaining of SRBD1, swollen metaphase cells were cytospun onto glass slides for 5 min at 1000 rpm, fixed (3% formaldehyde, 0.25% Triton X-100 in 1x PBS) and blocked with 1% BSA in KCM buffer (120 mM KCl, 20 mM NaCl, 10 mM Tris pH 8, 0.5 mM EDTA, 0.1% Triton X-100). Samples were incubated with GFP antibody (1:100; Proteintech, pabg1) diluted in blocking buffer at 4 °C overnight, followed by two washes in KCM buffer. After incubation with fluorescently labelled secondary antibody, samples were washed with KCM buffer, stained with DAPI (Sigma, D9542), and mounted with Vectashield mounting media (Vector Labs, H19002). For immunostaining of mitotic scaffold proteins, cells were synchronized in G2 phase as described above, and treated with DMSO or 1 µM 5-Ph-IAA (Tocris Bioscience, 739210) for 1 h. Cells were washed four times with pre-equilibrated media and released into 0.1 µg/ml demecolcine for 35–45 min. Mitotic cells were collected by shake-off, swollen as described above, fixed with methanol:acetic acid (3:1) for 2 h on ice and dropped onto pre-hydrated glass slides. Slides were re-hydrated in 1x PBS for 10 min at room temperature, blocked at room temperature for 2 h, and incubated in SMC2 (1:100; William Earnshaw or abcam, ab10412) and Topo IIα (1:100; Santa Cruz Biotechnology, sc-166934) primary antibodies at 4 °C overnight. Slides were washed three times with PBS, incubated with secondary antibodies for 1 h at room temperature, washed again with PBS, and mounted using ProLong Gold containing DAPI (ThermoFisher Scientific, P36941). Images were obtained using a 40X oil objective (Nikon Eclipse Ti) and mitotic chromosome integrated intensity was quantified using CellProfiler.

## Flow cytometry

Cells were fixed with ice cold 70% ethanol, permeabilized with 0.25% Triton X-100 on ice, and treated with propidium iodide and RNase A. Where indicated, cells were additionally stained with pS10 histone H3 antibody (1:50; Cell Signaling Technology, 3377). A nocodazole-treated control sample was included to ensure this staining measured mitotic cells. Samples were analyzed on an Attune flow cytometer.

## Cell proliferation assays

Cells were plated for clonogenic survival assays 48 h after siRNA transfection or plated and subsequently treated with DMSO or 1 µM 5 Ph-IAA (Tocris Bioscience, 739210). Colonies were allowed to grow for 10–14 days prior to staining with methylene blue (48% methanol, 2% methylene blue, 50% water). For the short-term viability assays, cells were plated 72 h after siRNA transfection and grown for an additional 72 h before measuring metabolic activity with alamarBlue (Invitrogen, DAL1100). All viability measurements are displayed as a percentage of the control and represent at least two biological replicates containing three technical replicates each.

## Live cell imaging

Cells expressing H2B-mCherry were generated by retroviral infection. Cells were plated on a glass bottom chambered coverslip (ibidi USA, 80287) or 35 mm glass bottom dish (MatTek Corporation, P35G-1.5-14-C), and synchronized in G2 phase with 6 µM Ro-3306 (ApexBio, A8885) for 16 h, where indicated. G2 synchronized cells were treated for 1 h with DMSO or 1 µM 5 Ph-IAA (Tocris Bioscience, 739210),

washed 5–7 times with equilibrated media, and released into McCoys 5 A media containing BackDrop Background Suppressor (Invitrogen, B10512) immediately before imaging. Asynchronously growing cells were treated with DMSO or 1 µM 5 Ph-IAA for 1 h immediately prior to imaging. For prometaphase degradation or delay, cells were first synchronized in G2 phase with 6 µM Ro-3306 (16 h; ApexBio A8885), washed 5–7 times, and released into equilibrated media containing 100 ng/ml nocodazole for 2 h. DMSO or 1 µM 5 Ph-IAA was added during the second hour of nocodazole treatment (prometaphase degradation) or the last hour of G2 arrest before release into nocodazole, cells were washed 7–8 times and released into equilibrated media immediately before imaging. For visualization of tubulin dynamics, CellLight Tubulin-GFP (ThermoFisher Scientific, C10613) was added to cells for 20–24 h, with 1 µM 5-Ph-IAA (Tocris Bioscience, 739210) added during the final 1 h, and asynchronously growing cells were imaged. Time-lapse imaging was performed in a heated incubation chamber (37 °C) with controlled humidity and 5% $CO_2$, using a Plan Apo Lambda 60x/1.40 NA WD 0.13 mm oil objective mounted on a Yokogawa CSU-X1 spinning disk field scanning confocal system with high-speed piezo [z] stage and four laser lines (405 nm, 488 nm, 561 nm, 647 nm). Images were acquired using an Andor DU-897 EMCCD camera, and 12-16 z-planes with 1 µm separation were collected every 2 min (synchronized cells) or 3-4 min (asynchronous). Cells were manually scored for the mitotic defects indicated.

## Metaphase chromosome length analysis

Cells were synchronized in G2 phase with 6 µM Ro-3306 for 6 h (ApexBio, A8885), and DMSO or 1 µM 5-Ph-IAA (Tocris Bioscience, 739210) was added for the last 1 h. Cells were washed three times with pre-equilibrated media, and released into media containing 0.1 µg/ml demecolcine (MilliporeSigma, D1925) for 2 h. Metaphase cells were harvested by mitotic shake-off, pelleted by centrifugation, and swollen by incubation in pre-warmed 75 mM KCl for 15 min at 37 °C. Swollen cells were collected by centrifugation, fixed with methanol:acetic acid (3:1) for 1 h on ice, and dropped onto pre-hydrated glass slides. Chromosomes were stained with DAPI (Vectashield; Vector Labs, H1200) and imaged using a 40X oil objective (Nikon Eclipse Ti). Chromosome length was measured using Nikon Elements software.

## RNA sequencing

RNA was collected from asynchronous U2OS cells 24 or 48 h after transfection with non-targeting or SRBD1 siRNAs using the Aurum total RNA mini kit (Bio-Rad). The library was prepared with the NEB stranded mRNA kit according to the manufacturer's instructions, sequencing was performed on the Illumina NovaSeq 6000, and 30 million reads were generated for each sample. Reads were trimmed to remove adaptor sequences and aligned to hg38 using HiSat2. FeatureCounts was used to count the number of mapped reads to each gene and differential gene expression analysis was performed with edgeR.

## Statistics and reproducibility

All statistical analyses were completed using Prism 10 (GraphPad), with details of the individual statistical tests indicated in the figure legends. No statistical methods were used to estimate sample size or to include/exclude samples. Multiple siRNAs, clones, and approaches were analyzed to confirm results were not caused by off-target effects or clonal variations. All experiments were performed at least twice (unless otherwise indicated in the figure legend). Graphs display all data points analyzed across replicate experiments along with the mean of each replicate experiment.

## Reporting summary

Further information on research design is available in the Nature Portfolio Reporting Summary linked to this article.

## Data availability

The RNA-seq data generated in this study is available in the GEO repository under accession number GSE283162. The previously published proteomics data that was reanalyzed in this study is available via PRIDE accession number PXD020914. All other data generated during this study are included in the article and its supplementary information files, including the source data file provided with this paper. Source data are provided with this paper.

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

## Acknowledgements

This research was supported by NIH grants R01CA239161 and R01GM116616 to D.C. Additional support came from the Vanderbilt-Ingram Cancer Center and the Breast Cancer Research Foundation. We thank William Earnshaw for sharing the SMC2 antibody, Ian Hickson for reagent recommendations, as well as Todd Stukenberg and Frank Mason for useful discussions.

## Author contributions

D.C., S.R.W., and C.A.L. conceived of the project. D.C. supervised the project. C.A.L., S.R.W., R.B., and R.Z. performed most of the experimentation. Y.H. generated the SRBD1-degron cells under the supervision of M.T.K.

## Competing interests

The authors declare no competing interests.
