## [Transparent Peer Review file · Nature Communications]

SRBD1 facilitates chromosome segregation by promoting topoisomerase II α localization to mitotic chromosomes

Corresponding Author: Dr David Cortez

Version 0:

Reviewer comments:

Reviewer #1

(Remarks to the Author)

The S1 RNA-binding domain containing protein 1 (SRBD1) has not yet been characterized properly. In this manuscript, the authors demonstrate that SRBD1 binds ss and ds DNA and RNA, histones, and with newly synthesized chromatin during DNA replication. The authors find that SRBD1 is essential for cell survival and that its inactivation produces DNA damage, micronuclei, and chromatin bridges. Reducing TOP2A activity or depleting condensin II attenuates the phenotypes of SRBD1 inhibition. Selective degradation of SRBD1 immediately before mitotic entry causes a complete anaphase failure, while it was observed to be not required to promote cell division after chromosome condensation. Anaphase failure was accompanied by defects in properly condensing and decatenating chromosomes. These data identify SRBD1 as a new factor required for chromosome segregation. SRBD1, by a yet unknown mechanism, seems to contribute to genome maintenance during early mitosis by aiding the generation of sister chromatids that are primed for anaphase segregation.

Overall, while this reviewer feels that the work presented is novel, the work falls short in providing mechanistic details of how SRBD1 contributes to chromosome organization in early mitosis. Many of the directions mentioned in the discussion are easily testable, at least to provide an initial insight into the mechanism of SRBD1. Additionally, there is a need for a more in-depth characterization of the phenotype and some of the comments below will help accomplish that goal.

- i. Figure S1: For S1A, please use different colors for the overlapped structures as it is easy to see. Better description of results related to S2A and 2B is needed in the main text and the legends in addition to what has been provided the methods. For example, what do orange and green colors denote? GNP NAT1 seems to be spelt wrong on one occasion.
- ii. Mention in line 113-114 that Fig. 2C and 2D are 48 hrs and Sup Fig. 2C and 2D are 72 hrs. What is the significance of doing this at two different time points?
- iii. This reviewer feels that it is a must to show in the increase in γ -H2AX in Fig. 3 and Suppl. Fig. 3 by immunofluorescence.
- iv. The results in Fig. 3D-F need to be described properly in the main text rather than just citing data in Suppl fig. 4. For example, it would be good to add more details pertaining to PICH and UBF here. What is the purpose of CENP-B staining in Suppl. Fig. 4? All it does is to say whether the bridges contain centromeres or not. This is useful to identify whole chromosome mis-segregation and not bridges.
- v. This reviewer has a major issue with using G2 synchronized cells to study phenotypes of early mitosis as CDK1 is critical for some many processes at G2-M transition and early mitosis. Double Thymidine release might be a better method to use. I don't think this statement "these phenotypes were not dependent on the CDK1 inhibition used to achieve G2 synchronization since degradation of SRBD1 in asynchronous cells also caused anaphase failure and chromatin bridges" is completely valid as many things could lead to chromatin bridges in asynchronous cells lacking SRBD1. Please include all details of the cell synchronization methods used in the methods section.
- vi. The data presented in Sup fig 7A is not at all convincing to state that spindles are overall normal. This is important because defects of the spindle later in mitosis could also cause bridges and mis-segregation. Better data is needed here. Chromosome labelling is needed in Fig. S7B, especially to discern what is happening in prophase and prometaphase. Even antibody staining in fixed cells would be ok if the authors can provide this data.

vii. It is critical to show that SRBD1 staining is lost in Fig. 6C. Blots to show knockdown to TOP2A is also required. Please reiterate for the purpose of the readers, why rescue of anaphase failure is important in the context of TOP2A and condensin knockdown experiments. Based on what I see in Fig. 6F, I would not claim that CAP-H2 rescues anything. Things look quite drastic in this case as well. Some rewording of the results section in this regard is necessary.

viii. In general, the results shown by fixed staining of the altered chromosome morphology in SRBD1-inhibited cells is not obvious to see in the many live images of chromosome segregation provided in the manuscript. Could the authors please comment about this?

Minor comments:

i. Line 34: Text unclear about which two processes. I am assuming genome duplication and chromosome segregation. Maybe, starting a new paragraph is not needed here.

ii. Fig. 2E: It might be better to point out the DNA content here for ease of readers outside of cell cycle field.

iii. It might be better to have insets in fig 4C to show lagging chromosomes or other defects observed.

Reviewer #2

(Remarks to the Author)

In this manuscript, the authors reported the identification of S1 RNA Binding Domain Containing Protein 1 (SRBD1) that was enriched at replication forks. They showed that SRBD1 associated with histones and its downregulation led to increased DNA damage signaling. Interestingly, SRBD1 depleted cells appeared to be arrested at G2 phase and this phenotype was partially rescued by TOP2A downregulation. The authors used an auxin-inducible degradation system and showed that cells with SRBD1 depletion also displayed mitotic chromosomal segregation defects, which indicates a potential role of SRBD1 in early mitosis.

The identification of SRBD1 as a component of mitotic chromosome and its role in genome maintenance is potentially interesting. However, the current study is premature and does not address key questions regarding the regulation and function of SRBD1. For example, is SRBD1 cell cycle regulated? is SRBD1 a chromatin binding protein? Does chromatin-association of SRBD1 require its ability to bind to DNA, RNA, or histones? What are the structural functions of SRBD1 on mitotic chromosomes? Are there proteins co-depleted with SRBD1 when SRBD1 is degraded? TOP2A and condensin II also act in early mitosis. Why knockdown of these proteins are able to partially rescue but not aggravate the mitotic defects?

Reviewer #3

(Remarks to the Author)

In the present study, Lovejoy et al. identified a previously uncharacterized protein called SRBD1, establishing its identity as a histone and nucleic acid binder. Underlining the importance of SRBD1 is its essential role for cell viability. SRBD1 is enriched at nascent DNA replication but plays a major role during mitosis. Specifically, SRBD1 is required in early mitosis, ensuring proper anaphase and, consequently, facilitating cell division. To unravel SRBD1's role in mitotic phases, the authors employed an AID-degron system, enabling the rapid degradation of the protein. This approach, coupled with live cell imaging, provided clear data confirming its role in early mitosis.

The data are overall very convincing, except for the rescue observed upon TOP2A or condensin depletion, which should be further validated. To enhance the robustness of these findings, the authors could consider combining TOP2 and Condensin depletion. To decrease the impact that depleting these factors could have on cell growth, the authors could use the inducible degradation to shortly deplete those proteins.

Prolonged arrest in prometaphase also contribute to rescue the phenotypes observed upon SRBD1 depletion. This suggest that another pathway/protein could at least partially backup SRBD1 function. This could be elaborated more in the discussion section. Moreover, a combination of prolonged mitosis and either TOP2A or Condensin depletion should give a better rescue phenotype.

In summary, the manuscript should be considered for publication in Nature communications after appropriate revisions along the lines suggested below.

- Just one representative graph is not suitable for immunofluorescence quantification (gH2AX, RPA, etc.) and mitotic abnormalities count. Although this is definitely acceptable for western blot, microscopy data from triplicate experiments should be represented all in the same plot. This can be achieved by plotting all the data points or, if this is not doable for the intensity quantification, the mean/median of each experiment can be plotted (either normalized or not to the untreated control sample). Only in this case, proper statistical tests could be applied.
- In general, the number of replicates should be included in each figure legend along with the name of the applied statistical test.
- The authors should add more details explaining how cells are categorized as G1/S/G2/M for TOP2A nuclear intensity analysis in Fig. 2B.
- In Fig 2E, the 4N cells should be quantified and plotted as duplicates/triplicates.

- The authors report a striking increase in TOP2A nuclear intensity in G2/M cells. However, they could not detect any increase in TOP2A covalent complexes by RADAR assay. Cells depleted for SRBD1 have increased number of G2/M cells. How can the authors rule out that this is not due to an increased number of G2/M cells, considering that TOP2A is also increased in G2/M in normal conditions? To explain the apparent discrepancy between immunofluorescence and RADAR data, the authors should perform the RADAR assay in fractionated cells.
- The absence of SRBD1 leads to a mild increase of gH2Ax intensity in asynchronous cells. It would be interesting to specifically look at gH2Ax in mitotic cells, after SRBD1 depletion by siRNAs or AID rapid degradation. This could help understating if the damage is due to SRBD1 role in mitosis or during replication.
- The authors prove that the SRBD1 depletion has major effects in mitosis. This was extensively covered. However, SRBD1 strongly associates with nascent DNA and its depletion leads to increased RPA and ssDNA. Does SRBD1 have a role in DNA replication fork progression? Do cells experience a longer S-phase?
- The AID degenon is used to convincingly prove SRBD1 role in mitosis. This could also be applied the other way around and deplete SRBD1 only in S-phase to look at the consequences in mitosis.
- Statistics are missing in Fig. 6.

Version 1:

Reviewer comments:

Reviewer #1

(Remarks to the Author)

The authors have addressed most of my concerns, but I see some issues still persist. A general comment is that: The authors describe the new data in support of the new mechanical insights obtained in the rebuttal, but it would have been better if they provided a lay paragraph description on what precisely the new mechanical insights were and why the findings were important. Other specific points are:

1. I am sorry, I still have a major issue with the live-cell data shown to claim that mitotic spindles are normal overall. Maybe the issue is that they do not have a control movie to show here. It is possible that the spindles look more or less normal until metaphase, but considering the drastic chromosomal phenotype and based on what data is shown in the live imaging experiments, what I see is that there is a major spindle phenotype later in mitosis post-metaphase. Maybe I am focusing on the overall general microtubule structure. Maybe the authors can use specific language in the text to temper down the claims? Maybe show still pictures of anaphase/telophase spindles and/or exclude the live-imaging data? This reviewer has no difficulty in believing that spindle in later stages of mitosis could be abnormal if there is a major issue with chromosome compaction or in the case of major mis-segregation events/entanglements and bridges.
2. I think one of the other reviewers also alluded to this in the initial review; but what I wanted with regard to the data in the new Figure 3B and S2C is for them to simply show immunofluorescence pictures of the data used for this quantification and not just the quantified data. I do not see why this should be a problem.
3. I still do not see textual details about the significance of using PICH and UBF included in the manuscript. I thought this would benefit the general readers. I was not just asking about details of the reagents used in this regard.
4. Minor: In new Figure 7E, since the authors have already quantified this data, it might be better to increase the overall brightness of the figure as this look quite dim (even in the cases where one is supposed to see bright TOP2A).
5. Minor: I did not notice this in the previous round of submission, but I am curious to know if the bottom most band in the top blot of Figure 3A which runs very close to SRDB1 in the case of mAC-SRBD1 (c1) is leaky endogenous expression or if this is possibly some other protein?

Reviewer #2

(Remarks to the Author)

The revised manuscript provided some additional data to further define the mechanisms and/or functions of SRBD1 in mitosis. The key data is that although SRBD1 and CAP-D3 depletion each reduced TOP2A chromatin association, double depletion of these proteins somehow increased TOP2A chromatin loading, which the authors hypothesized to be the mechanism underlying the mitotic phenotypes observed in cells with SRBD1 depletion.

These data seem to disagree with their previous observation that depletion of TOP2A was also able to partially rescue the mitotic phenotypes observed in cells with SRBD1 depletion. They have now removed these data in the revised manuscript.

The concern is that CAP-D3 depletion by siRNAs took at least two days. It remains possible that the rescue phenotypes observed in these double depletion experiments may be due to some indirect effects on cell proliferation and/or mitotic progression. Since the authors would like to use this unique genetic interaction to draw their main conclusion, i.e. SRBD1 facilitates chromosome segregation by promoting TOP2A localization to mitotic chromosomes, I agree with the other reviewer that inducible co-depletion of these two proteins may be needed to further verify this interesting genetic interaction.

Reviewer #3

(Remarks to the Author)

The authors have made thorough revisions and addressed all my comments comprehensively. I believe that the revised version of the manuscript has significantly improved. The revisions have strengthened the clarity and rigor of the work, and I am satisfied with the changes made. I have no further comments or suggestions at this time.

In my opinion, the manuscript is now suitable for publication in Nature Communications.

Version 2:

Reviewer comments:

Reviewer #1

(Remarks to the Author)

I am satisfied with the revisions. It would be good to include relevant details from the authors response to my 1st query (lay paragraph description of significance) in the discussion section of the manuscript if they have not already done so.

Reviewer #2

(Remarks to the Author)

The revised manuscript addressed some of my concerns. Unfortunately the authors do not have inducible co-depletion cells to further support their working hypothesis. Nevertheless, the authors performed siRNA mediated depletion of CAP-H2 and CAP-D3 in their SRBD1 inducible cell line. They should be able to comment on whether CAP-H2 or CAH-D3 depletion is able to rescue cell lethality caused by SRBD1 depletion.

Reviewer #1 (Remarks to the Author):

The S1 RNA-binding domain containing protein 1 (SRBD1) has not yet been characterized properly. In this manuscript, the authors demonstrate that SRBD1 binds ss and ds DNA and RNA, histones, and with newly synthesized chromatin during DNA replication. The authors find that SRBD1 is essential for cell survival and that its inactivation produces DNA damage, micronuclei, and chromatin bridges. Reducing TOP2A activity or depleting condensin II attenuates the phenotypes of SRBD1 inhibition. Selective degradation of SRBD1 immediately before mitotic entry causes a complete anaphase failure, while it was observed to be not required to promote cell division after chromosome condensation. Anaphase failure was accompanied by defects in properly condensing and decatenating chromosomes. These data identify SRBD1 as a new factor required for chromosome segregation. SRBD1, by a yet unknown mechanism, seems to contribute to genome maintenance during early mitosis by aiding the generation of sister chromatids that are primed for anaphase segregation.

Overall, while this reviewer feels that the work presented is novel, the work falls short in providing mechanistic details of how SRBD1 contributes to chromosome organization in early mitosis. Many of the directions mentioned in the discussion are easily testable, at least to provide an initial insight into the mechanism of SRBD1.

We appreciate that the reviewer recognizes the novelty of our findings on SRBD1. As recommended, we focused our efforts on providing mechanistic insights into the mitotic functions of SRBD1. The revised manuscript contains 18 new figure panels including the following new, key results that are relevant to the mechanism by which SRBD1 regulates the organization of mitotic chromosomes:

1. We found that SRBD1 is required for the proper chromosomal localization of TOP2A during mitosis (new Figures 6d, e and Figures S7d, e in the revised manuscript).

TOP2A and condensin complexes are localized along the central axes of mitotic chromosomes to promote chromosome individualization and compaction^{1, 2, 3, 4, 5, 6}. We found that acute inactivation of SRBD1 using the SRBD1 degron causes a striking reduction in the amount of TOP2A on mitotic chromosomes. Very little TOP2A is visible along the chromosome arms and the amount at centromeres is also reduced (new Fig. 6d, e). In contrast, we did not observe any overt defect in the localization or abundance of SMC2, a subunit of both condensin complexes (new Fig. 6d and new Fig. S7d). Insufficient TOP2A activity along the chromosome arms in SRBD1-deficient cells likely explains the anaphase bridges (Fig. 3e-g), the defects in chromosome compaction (Fig. 6b, c), and the extensive sister chromatid entanglements (Fig. 4) caused by SRBD1 inactivation⁷.

2. Loss of condensin II improves TOP2A localization to mitotic chromosomes and rescues the chromosome segregation defects in SRBD1-deficient cells (new Fig. 7d, e and new Fig. S8b, c in the revised manuscript).

The anaphase failure phenotype of SRBD1-deficient cells is strongly alleviated by inactivating condensin II (Fig. 7a, b). To understand if this improvement was related to TOP2A regulation, we examined TOP2A localization on mitotic chromosomes in SRBD1-deficient cells transfected with CAP-D3 siRNA to inactivate condensin II. While depletion of CAP-D3 or SRBD1 degradation individually caused a dramatic reduction in TOP2A association with metaphase chromosomes, the combined loss of CAP-D3 and SRBD1 largely rescued TOP2A localization to mitotic chromosome arms (new Fig. 7d, e). Explanations for synthetic rescue in other settings include rebalancing of an activity or pathway, activation of compensatory pathways, or restoration of homeostasis. Further studies will be needed to understand the mechanism in this case.

Nonetheless, this result provides an explanation for how condensin II inactivation greatly improves the chromosome segregation phenotypes of SRBD1-deficient cells, and further argues that SRBD1 is critical for proper regulation of TOP2A to decatenate mitotic chromosomes.

Additionally, there is a need for a more in-depth characterization of the phenotype and some of the comments below will help accomplish that goal.

i. Figure S1: For S1A, please use different colors for the overlapped structures as it is easy to see. Better description of results related to S2A and 2B [S1B and S1C] is needed in the main text and the legends in addition to what has been provided in the methods. For example, what do orange and green colors denote? GNPAT1 seems to be spelt wrong on one occasion.

We changed the colors in Fig. S1a, as recommended. We also corrected the spelling error and added a better description of the RNAseq results for Fig. S1b and S1c. The MA plots show the \log_2 fold change for each gene plotted against the average \log_2 counts per million. The few genes showing significant decreases or increases in expression after siRNA depletion of SRBD1 are listed and colored orange or green, respectively, in the tables below the MA plots. Notably, SRBD1 is the most downregulated gene at both 24 and 48 hours after siRNA transfection. The very limited number of gene expression changes suggested SRBD1 does not likely regulate transcription.

ii. Mention in line 113-114 that Fig. 2C and 2D are 48 hrs and Sup Fig. 2C and 2D are 72 hrs. What is the significance of doing this at two different time points?

We performed the analysis at two time points after siRNA knockdown to determine if the phenotypic effects accumulate over time. In the revised manuscript, we removed the data for the 72-hour time point since significant cell death was observed at 72 hours after siRNA transfection, and we added a third replicate of the siRNA experiments analyzing DNA damage signaling and single-stranded DNA at the 48-hour time point (new Fig. 2a-d).

iii. This reviewer feels that it is a must to show the increase in γ -H2AX in Fig. 3 and Suppl. Fig. 3 by immunofluorescence.

We examined γ H2AX by immunofluorescence in both SRBD1 degron clones, as requested (new Fig. 3b and new Fig. S2c). As expected, a significant increase in γ H2AX can be observed by 4 hours after SRBD1 degradation, and possibly earlier in one clone, with larger increases by 16 and 24 hours.

iv. The results in Fig. 3D-F [Fig. 3e-g] need to be described properly in the main text rather than just citing data in Suppl fig. 4 [Supplemental Fig. 3]. For example, it would be good to add more details pertaining to PICH and UBF here. What is the purpose of CENP-B staining in Suppl. Fig. 4? All it does is to say whether the bridges contain centromeres or not. This is useful to identify whole chromosome mis-segregation and not bridges.

We apologize for not providing a more detailed description of these experiments. We were interested in understanding the type and source of the anaphase bridges observed in SRBD1-deficient cells. We found that most chromatin bridges did not contain CENPB or UBF foci, suggesting that the sister chromatid intertwinings did not originate primarily from centromeric or ribosomal DNA. Extensive anaphase chromatin bridges that do not originate from specific genomic loci would be consistent with the massive sister chromatid entanglements visualized during live-cell imaging and the defective TOP2A localization along

metaphase chromosome arms following SRBD1 degradation. We have amended the main text and legends for Fig. 3 and Supplemental Fig. 3 to provide a better description of this data.

v. This reviewer has a major issue with using G2 synchronized cells to study phenotypes of early mitosis as CDK1 is critical for some many processes at G2-M transition and early mitosis. Double Thymidine release might be a better method to use. I don't think this statement "these phenotypes were not dependent on the CDK1 inhibition used to achieve G2 synchronization since degradation of SRBD1 in asynchronous cells also caused anaphase failure and chromatin bridges" is completely valid as many things could lead to chromatin bridges in asynchronous cells lacking SRBD1. Please include all details of the cell synchronization methods used in the methods section.

We appreciate the reviewer's concern about the use of a CDK1 inhibitor to synchronize cells and apologize for not clearly explaining how we did the "asynchronous" cell experiments. Inactivation of SRBD1 after S phase was critical to analyze the mitotic functions of SRBD1 independently of its effects in S-phase. To ensure the phenotypes we observed with CDK1 inhibition were not a synthetic effect of the synchronization and SRBD1 degradation, we also performed the live imaging experiments without synchronization. By limiting SRBD1 degradation to 1 hour and then only scoring cells that entered mitosis within the next hour, we ensured these experiments examined cells in which SRBD1 was inactivated after they had already completed S-phase. Since cells spend approximately 3-4 hours in G2 phase of the cell cycle, this procedure effectively only analyzed cells in which SRBD1 was inactivated in late G2 phase cells prior to entering mitosis. Anaphase failure remains the predominant phenotype after SRBD1 degradation in the absence of CDK inhibition (Fig. 4c). In the revised manuscript, we also include additional live imaging experiments without CDK1 inhibition (e.g. Fig. 7b, c- orange data points). We considered using a double-thymidine block, as suggested; however, our procedure in asynchronous cells confirmed that the mitotic phenotypes observed in SRBD1-deficient cells are not due to S-phase problems or synthetic effects caused by synchronization, and thymidine treatment induces a form of replication stress that we wanted to avoid. We changed the text to more accurately describe the conditions of each experiment. We also ensured that the methods section contains details of the synchronization used in each experiment.

vi. The data presented in Sup fig 7A [Supplementary Fig. 6c] is not at all convincing to state that spindles are overall normal. This is important because defects of the spindle later in mitosis could also cause bridges and mis-segregation. Better data is needed here. Chromosome labelling is needed in Fig. S7B [Supplementary Fig. 7a], especially to discern what is happening in prophase and prometaphase. Even antibody staining in fixed cells would be ok if the authors can provide this data.

As recommended, the revised manuscript contains a more thorough analysis of the mitotic spindle. We continue to find no evidence of major defects when SRBD1 is inactivated. We visualized the spindle in fixed mitotic cells to evaluate spindle aberrations in both SRBD1 degron clones. We observed no instances of monopolar spindles and rarely observed multipolar anaphases in either control or SRBD1-deficient cells. We also found no evidence of unfocused or splayed spindle poles. Furthermore, spindle length measurements from metaphase cells revealed no change following SRBD1 degradation (new Fig. S6a, b). We also added still images from the live-cell imaging experiments that better display SRBD1 localization during each mitotic stage (new Fig. S7a). Since we have not observed any significant mitotic spindle defects by live imaging or on fixed cells, it seems unlikely that spindle problems are the cause of the massive bridging and mis-segregation observed after SRBD1 degradation.

vii. It is critical to show that SRBD1 staining is lost in Fig. 6C. Blots to show knockdown to TOP2A is also required. Please reiterate for the purpose of the readers, why rescue of anaphase failure is important

in the context of TOP2A and condensin knockdown experiments. Based on what I see in Fig. 6F, I would not claim that CAP-H2 rescues anything. Things look quite drastic in this case as well. Some rewording of the results section in this regard is necessary.

Our immunoblots have consistently shown little or no detectable SRBD1 after 5-Ph-IAA treatment, and the immunostaining of cytospin spreads also showed loss of almost all SRBD1 staining. Unfortunately, we are unable to stain for SRBD1 on the same metaphase spreads from which chromosome length was measured. Neither the SRBD1 or GFP antibodies we tested were compatible with methanol/acetic acid fixation, a necessary step to spread metaphase chromosomes using gravity (and humidity).

Regarding the condensin depletion, we agree with the reviewer that “rescues” is not the most appropriate word choice to describe these phenotypes. Rather, condensin II depletion significantly reduces the fraction of SRBD1-deficient cells that fail anaphase. This reduction in mitotic failure produces a corresponding increase in the less severe anaphase chromatin bridges (Fig. 7c) and the cells are largely able to segregate the sister chromatids, often retaining only a thin chromatin bridge between the two daughter cells (Fig. 7a, siCAP-H2 +5-Ph-IAA). Anaphase failure was reduced from 60% in SRBD1-deficient cells to 30% when combined with CAP-H2 depletion and 20% with CAP-D3 depletion. The combined inactivation of CAP-D3 and SRBD1 even yielded some anaphases with no apparent defects. These phenotypic results are consistent with our new data that shows CAP-D3 knockdown mostly restores proper TOP2A localization to the mitotic chromosomes in SRBD1-deficient cells (new Fig. 7d, e). We modified the text to indicate that condensin II depletion improves, but does not fully rescue, the mitotic defects caused by SRBD1 inactivation.

The original Fig. S2a showed the efficiency of TOP2A depletion after siRNA transfection. However, this panel and the associated experiments have been removed from the revised manuscript as detailed in our response to reviewer 3, below.

viii. In general, the results shown by fixed staining of the altered chromosome morphology in SRBD1-inhibited cells is not obvious to see in the many live images of chromosome segregation provided in the manuscript. Could the authors please comment about this?

The average change in chromosome length after SRBD1 degradation (and 2 hr in demecolcine) is about 1 μm when measured on fixed chromosome spreads. We agree that it is difficult to visualize this on images from live cells. The chromosomes in the live cells are in 3-dimensional space and not separated from neighbors, in contrast to the flattened and individualized chromosomes spread from fixed metaphase cells. It is not possible to accurately quantitate individual chromosome lengths from the live cell images.

Minor comments:

i. Line 34: Text unclear about which two processes. I am assuming genome duplication and chromosome segregation. Maybe, starting a new paragraph is not needed here.

We apologize for the confusion. The reviewer is correct; “both processes” refers to genome duplication and chromosome segregation. The sentence has been changed in the revised text.

ii. Fig. 2E: It might be better to point out the DNA content here for ease of readers outside of cell cycle field.

Thank you for the suggestion. DNA content is now indicated in Fig. 2e.

iii. It might be better to have insets in fig 4C to show lagging chromosomes or other defects observed.

We identified the lagging chromosomes by an arrow in the appropriate panels in the revised manuscript (now Fig. 4a).

Reviewer #2 (Remarks to the Author):

In this manuscript, the authors reported the identification of S1 RNA Binding Domain Containing Protein 1 (SRBD1) that was enriched at replication forks. They showed that SRBD1 associated with histones and its downregulation led to increased DNA damage signaling. Interestingly, SRBD1 depleted cells appeared to be arrested at G2 phase and this phenotype was partially rescued by TOP2A downregulation. The authors used an auxin-inducible degradation system and showed that cells with SRBD1 depletion also displayed mitotic chromosomal segregation defects, which indicates a potential role of SRBD1 in early mitosis.

The identification of SRBD1 as a component of mitotic chromosome and its role in genome maintenance is potentially interesting. However, the current study is premature and does not address key questions regarding the regulation and function of SRBD1. For example, is SRBD1 cell cycle regulated? is SRBD1 a chromatin binding protein?

The revised manuscript contains 18 new figure panels further characterizing the regulation and function of SRBD1. To answer the reviewer's specific questions, we synchronized cells in each phase of the cell cycle and examined SRBD1 localization and abundance by subcellular fractionation. Immunoblots for SRBD1 show that protein expression does not appreciably change throughout the cell cycle, and additionally that SRBD1 is predominantly associated with chromatin independently of cell cycle stage. This new data has been added to the revised manuscript as new Figure 1b.

Does chromatin-association of SRBD1 require its ability to bind to DNA, RNA, or histones?

We agree with the reviewer that this is an important question, and we have begun to examine how SRBD1 associates with chromatin to determine whether its replication and/or mitotic functions are dependent on chromatin association. However, our preliminary results indicate there are likely multiple nucleic acid binding surfaces, and we have not yet been able to make the appropriate mutants that are stable and lose only nucleic acid or histone binding activities. The revised manuscript provides a substantial amount of new information on SRBD1, and completing these types of structure-function experiments will require extensive investigation that we believe is beyond the scope of the current study.

What are the structural functions of SRBD1 on mitotic chromosomes?

Our new data indicate the structural functions of SRBD1 on mitotic chromosomes are centered on TOP2A localization. In control cells, TOP2A localizes to the central axes of mitotic chromosomes, often with a visible enrichment at centromeres. In contrast, SRBD1-deficient cells show little staining of TOP2A along chromosome arms, with TOP2A largely restricted to centromeres and in less abundance than control cells (new Fig. 6d, e). This was not due to a difference in total TOP2A levels (new Fig. S7f). Furthermore, SRBD1 degradation produced no significant change in the localization of SMC2 (new Fig. 6d and new Fig. S7d).

Insufficient TOP2A activity could readily explain all mitotic defects we have observed following SRBD1 degradation, including the dramatic anaphase chromatin bridging (Fig. 3e-g), the extensive sister chromatid entanglements observed by live imaging (Fig. 4), as well as the defects in chromosome compaction (Fig. 6b, c). This new data provides key mechanistic insight on the mitotic functions of SRBD1.

We also attempted several additional experiments to examine the structural functions of SRBD1 on mitotic chromosomes that yielded negative results. For example, micrococcal nuclease assays did not reveal any large changes in the nucleosome organization of mitotic chromosomes (Reviewer Fig. 1). We did not include this negative data in the revised manuscript. Further experiments will be needed to understand how SRBD1 regulates TOP2A localization to mitotic chromosomes.

Reviewer Figure 1. SRBD1 degradation produces no overt changes in mitotic nucleosome organization. G2 phase synchronized SRBD1 degenon cells were treated with DMSO or 1 μ M 5-Ph-IAA for 1 hr and released into nocodazole for 35 min. Mitotic cells were collected and genomic DNA was digested with increasing amounts of micrococcal nuclease (MNase) for 5 min (a) or with 20 units of MNase for increasing amounts of time (b). Digested DNA was treated with RNase A and extracted with phenol/chloroform. DNA was recovered by ethanol precipitation, quantified, resolved by agarose gel electrophoresis, and visualized by ethidium bromide staining.

Are there proteins co-depleted with SRBD1 when SRBD1 is degraded?

We examined cell extracts by immunoblot and confirmed that neither SMC2 nor TOP2A are depleted by being in proximity of the E3 ligase targeting SRBD1 for degradation (new Fig. S7f). Additionally, SRBD1 does not appear to function as part of a multi-subunit complex since our SRBD1 purifications did not identify any stably associated proteins. Therefore, it also seems unlikely that SRBD1 degradation causes co-depletion of other critical proteins that could explain the phenotypic consequences of SRBD1 inactivation.

TOP2A and condensin II also act in early mitosis. Why knockdown of these proteins are able to partially rescue but not aggravate the mitotic defects?

Condensin II depletion produces a considerable improvement in the mitotic defects in SRBD1-deficient cells. Our analysis of TOP2A localization now provides an explanation for this partial rescue. Specifically, we find that TOP2A is not localized properly to mitotic chromosomes in SRBD1-deficient cells (new Fig. 6d, e). Although depletion of the condensin II subunit CAP-D3 by itself dramatically reduced TOP2A association

with metaphase chromosomes, the combined loss of CAP-D3 and SRBD1 improved TOP2A localization to metaphase chromosome arms to levels that are not significantly different from control cells (new Fig. 7d, e). This is a striking genetic result that explains why condensin II knockdown improves the most severe mitotic phenotypes associated with SRBD1 inactivation. Further studies will be needed to understand the underlying explanation of this genetic relationship.

We also found that partial inhibition of TOP2A catalytic activity with ICRF-187 yielded a very small (although statistically significant) improvement in the mitotic phenotypes of SRBD1-deficient cells (Reviewer Fig. 2). However, the reason for the minor improvement with ICRF-187 is not clear and it could be an indirect consequence of other changes in mitosis caused by partial inhibition of TOP2. We removed this observation from the manuscript since it does not add to the main conclusions.

Reviewer Figure 2. Partial inhibition of TOP2A with ICRF-187 modestly improves the anaphase failure phenotype of SRBD1-deficient cells. Quantitation of anaphase failure following G2 phase synchronization and treatment with DMSO or 1 μ M 5-Ph-IAA, followed by release into mitosis +/- TOP2A inhibition (DMSO or 200 nM ICRF-187). Graph displays the mean +/- SEM for two biological replicates of DMSO-treated G2 phase cells and mean +/- SD for three biological replicates with 5-Ph-IAA. Colored data points represent the mean of each biological replicate. P value was calculated using an unpaired t-test.

Reviewer #3 (Remarks to the Author):

In the present study, Lovejoy et al. identified a previously uncharacterized protein called SRBD1, establishing its identity as a histone and nucleic acid binder. Underlining the importance of SRBD1 is its essential role for cell viability. SRBD1 is enriched at nascent DNA replication but plays a major role during mitosis. Specifically, SRBD1 is required in early mitosis, ensuring proper anaphase and, consequently, facilitating cell division. To unravel SRBD1's role in mitotic phases, the authors employed an AID-degron system, enabling the rapid degradation of the protein. This approach, coupled with live cell imaging, provided clear data confirming its role in early mitosis.

The data are overall very convincing, except for the rescue observed upon TOP2A or condensin depletion, which should be further validated. To enhance the robustness of these findings, the authors could consider combining TOP2 and Condensin depletion.

We thank the reviewer for describing the data as clear and convincing. To enhance the robustness of our conclusions, we performed a third replicate of the live imaging after SRBD1 degradation with TOP2A

inhibition or condensin depletion. The reduction in mitotic failure with TOP2A inhibition remained statistically significant, but small (see Reviewer Fig. 2, above). We elected to remove this data from the manuscript since the differences are small and it isn't critical for our conclusions.

In contrast to TOP2 partial inhibition, condensin II depletion markedly and reproducibly decreased the mitotic failure phenotype caused by inactivating SRBD1 (new Fig. 7b). This figure has been updated in the revised manuscript to display the means of three independent experiments with P values. We also added new supplemental movies showing that the condensin II inactivation in the SRBD1-deficient cells can even produce normal looking anaphases, which were almost never observed when SRBD1 was inactivated alone.

Importantly, we also found that SRBD1 degradation impairs the mitotic localization of TOP2A (new Fig. 6d, e), and condensin II depletion combined with SRBD1 inactivation largely restores TOP2A localization to mitotic chromosomes (new Fig. 7d, e). These new results suggest that the reduction in anaphase failures in SRBD1-deficient cells after condensin II inactivation is due to an improvement in TOP2A localization, which promotes decatenation.

To decrease the impact that depleting these factors could have on cell growth, the authors could use the inducible degradation to shortly deplete those proteins.

We agree that examining mitotic defects after the inducible degradation of both SRBD1 and condensin would be ideal, but we have not yet been able to generate the appropriate dual degron cell lines required for this experiment. Nonetheless, to mitigate cell growth defects caused by depleting condensin subunits, our experiments analyzed mitotic defects 48 hours after siRNA transfection. We were able to confirm efficient depletion within this time frame and still observed no significant growth defects of these cells in culture. Importantly, milder chromosomal defects are observed when condensin is gradually lost over more than one cell cycle, possibly due to incomplete depletion or cellular adaptation^{8, 9, 10, 11, 12}. We hope the reviewer agrees that the new data added to the revised manuscript provides a compelling case that condensin depletion does improve the TOP2A localization and mitotic defects caused by SRBD1 inactivation.

Prolonged arrest in prometaphase also contribute to rescue the phenotypes observed upon SRBD1 depletion. This suggest that another pathway/protein could at least partially backup SRBD1 function. This could be elaborated more in the discussion section. Moreover, a combination of prolonged mitosis and either TOP2A or Condensin depletion should give a better rescue phenotype.

We appreciate the reviewer's suggestion to see if the mitotic defects can be further improved by combining different perturbations that individually promote a partial rescue. As the reviewer noted, the prometaphase delay alone already causes a striking improvement in mitosis following G2 phase degradation of SRBD1, with no anaphase failures observed and only a modest level of anaphase chromatin bridges remaining (Fig. 5b). This is consistent with the idea that SRBD1 causes reduced TOP2A localization to mitotic chromosomes, but providing more time for decatenation can rescue the SRBD1-deficient phenotypes. We tested whether combining the TOP2A inhibitor or condensin depletion with a prometaphase delay could further reduce the anaphase defects associated with SRBD1 inactivation, as recommended by the reviewer. The TOP2A inhibitor in combination with the prometaphase delay did not yield a consistent improvement and condensin II depletion slightly worsened the rescue phenotype, with a small but significant increase in anaphase chromatin bridges (Reviewer Fig. 3). Ultimately, more information about how each condition individually improves the mitotic defects of SRBD1-deficient cells is needed before we can understand any combined effect. Since these experiments did not provide interpretable results, we did not include them in the revised manuscript.

Reviewer Figure 3. Condensin II depletion combined with a mitotic delay worsens the anaphase defects of SRBD1-deficient cells. Cells were transfected with non-targeting (NT) or CAP-D3 siRNAs. G2 phase synchronized cells were treated with 1 μ M 5-Ph-IAA and released into 100 ng/ml nocodazole for 2 hr. After release from the nocodazole block, anaphase defects derived from live-cell imaging were analyzed. Graph displays the mean \pm SEM for two biological replicates. Colored data points represent the mean of each replicate and significance was determined using an unpaired t-test.

In summary, the manuscript should be considered for publication in Nature communications after appropriate revisions along the lines suggested below.

- **Just one representative graph is not suitable for immunofluorescence quantification (gH2AX, RPA, etc.) and mitotic abnormalities count. Although this is definitely acceptable for western blot, microscopy data from triplicate experiments should be represented all in the same plot. This can be achieved by plotting all the data points or, if this is not doable for the intensity quantification, the mean/median of each experiment can be plotted (either normalized or not to the untreated control sample). Only in this case, proper statistical tests could be applied.**

We have amended the graphs to represent all data points from replicate experiments with the means of each biological replicate shown on the same graph as larger, colored data points. The number of data points and statistical tests applied are indicated in each figure legend, and we compared the means of the replicate experiments (not all data points) to derive P values. We agree that this is a better representation of the data and thank the reviewer for the valuable suggestion.

- **In general, the number of replicates should be included in each figure legend along with the name of the applied statistical test.**

We apologize for any omissions of this information and have confirmed that all figure legends indicate the number of replicates, as well as the statistical test applied.

- **The authors should add more details explaining how cells are categorized as G1/S/G2/M for TOP2A nuclear intensity analysis in Fig. 2B.**

Our previous figure used DAPI intensity to delineate cell cycle stage by DNA content. However, additional experiments in the SRBD1 degron cells suggest the increase in TOP2A chromatin association after siRNA depletion of SRBD1 is an indirect effect of a cell cycle arrest or a failed cell division, as outlined in our response below. Therefore, we have removed this panel from the revised manuscript.

• In Fig 2E, the 4N cells should be quantified and plotted as duplicates/triplicates.

We revised the figure to include quantitation of cells with 4n and >4n DNA content from replicate experiments, as recommended.

• The authors report a striking increase in TOP2A nuclear intensity in G2/M cells. However, they could not detect any increase in TOP2A covalent complexes by RADAR assay. Cells depleted for SRBD1 have increased number of G2/M cells. How can the authors rule out that this is not due to an increased number of G2/M cells, considering that TOP2A is also increased in G2/M in normal conditions? To explain the apparent discrepancy between immunofluorescence and RADAR data, the authors should perform the RADAR assay in fractionated cells.

We thank the reviewer for this question. We analyzed TOP2A chromatin association by immunofluorescence in the SRBD1 degron cells and found that unlike siRNA depletion for two days, inactivation of SRBD1 in G1, S, or G2 phase arrested cells did not increase TOP2A nuclear intensity (Reviewer Fig. 4a). Cells arrested in S phase for SRBD1 degradation and then released and analyzed in G2 phase also failed to show an increase in TOP2A nuclear intensity (Reviewer Fig. 4b). This suggests that at least one cell division cycle is required for the increase in TOP2A on chromatin, and the interpretation of this increase is thus complicated by both the failed mitoses and G2 accumulation observed in SRBD1-deficient cells. Since the increase in TOP2A

Reviewer Figure 4. The insoluble TOP2A nuclear intensity in interphase cells is not elevated by SRBD1 loss. (a) Asynchronously growing SRBD1 degron cells were arrested in G1 (10 μ M lovastatin), S (3 mM hydroxyurea), or G2 phase (6 μ M Ro3306) of the cell cycle by 16 hr treatment with the indicated inhibitors. DMSO or 1 μ M 5-Ph-IAA was added and the cell cycle arrest maintained for an additional 8 hr. The amount of TOP2A in the detergent insoluble nuclear fraction was measured by immunofluorescence staining. (b) SRBD1 degron cells were arrested in S phase with 3 mM hydroxyurea for 16 hr. DMSO or 1 μ M 5-Ph-IAA was added for 1 hr and cells were released into 6 μ M Ro3306 for 16 hr. The amount of insoluble TOP2A in G2 phase cells was measured by immunofluorescence staining. (a-b) Gray data points represent the TOP2A nuclear intensity in one cell and all cells analyzed across 4 biological replicates are displayed (total $n \geq 8,977$). Black data points represent the mean nuclear intensity of each biological replicate, and black bars represent the mean of the replicate experiments. Significance was determined using an unpaired t-test comparing the means of the replicate experiments.

nuclear intensity after siRNA depletion of SRBD1 is an indirect effect that may be attributable (at least in part) to differences in cell cycle and DNA content, we have removed this panel from the revised manuscript.

• **The absence of SRBD1 leads to a mild increase of γ H2Ax intensity in asynchronous cells. It would be interesting to specifically look at γ H2Ax in mitotic cells, after SRBD1 depletion by siRNAs or AID rapid degradation. This could help understating if the damage is due to SRBD1 role in mitosis or during replication.**

We analyzed γ H2AX specifically in mitotic cells, as recommended. This required an evaluation by immunoblot because the poor adhesion of HCT116 cells to the plates used for high content imaging made it impossible to retain mitotic cells through the immunostaining procedure. The analysis of mitotic cells after SRBD1 degradation in G2 phase, with or without synchronization, showed no detectable change in γ H2AX by immunoblotting (Reviewer Fig. 5) suggesting damage arises in S-phase of SRBD1-deficient cells. This is consistent with the rapid increase in γ H2AX after SRBD1 degradation (new Fig. S2c) and the G2 cell cycle arrest that is relieved by DNA damage checkpoint inhibition (new Fig. 3d).

Reviewer Figure 5. Mitotic cells show no increase in γ H2AX immediately after G2 phase degradation of SRBD1. G2 phase cells were treated with DMSO or 1 μ M 5-Ph-IAA for 1 hr, with or without synchronization using a CDK1 inhibitor (Ro3306). Prometaphase cells were enriched by a short treatment with nocodazole (45 min) and mitotic cells were collected by shake-off. SRBD1 and γ H2AX were analyzed by immunoblotting.

• **The authors prove that the SRBD1 depletion has major effects in mitosis. This was extensively covered. However, SRBD1 strongly associates with nascent DNA and its depletion leads to increased RPA and ssDNA. Does SRBD1 have a role in DNA replication fork progression? Do cells experience a longer S-phase?**

We found no significant change in replication fork progression by molecular combing (Reviewer Fig. 6). Since fork progression is not impacted by SRBD1 degradation, yet cells are arrested in G2 phase by DNA damage checkpoint signaling, SRBD1 loss may be impacting chromatin architecture behind replication forks in a way that produces elevated single-stranded or damaged DNA. This negative data was not added to the manuscript.

Reviewer Figure 6. Acute inactivation of SRBD1 does not slow replication fork progression. Parental or SRBD1 degron cells were treated with DMSO or 1 μ M 5-Ph-IAA (1 hr for means colored orange and blue; 3 hr for means colored green) and labeled with CldU followed by IdU (20 min each) before DNA combing. Gray data points represent individual IdU fiber lengths, with all fiber lengths from 2-3 biological replicates shown. Colored data points represent the mean of each biological replicate and black bars represent the mean of the replicate experiments. P values were calculated using an ordinary one-way ANOVA with Sidak's multiple comparisons test.

• **The AID degron is used to convincingly prove SRBD1 role in mitosis. This could also be applied the other way around and deplete SRBD1 only in S-phase to look at the consequences in mitosis.**

Our data indicate that SRBD1 degradation in S-phase activates a DNA damage checkpoint that arrests cells in G2, precluding an analysis of the mitotic effects of S-phase depletion (at least in checkpoint proficient cells). We agree with the reviewer that it will be interesting to do a more thorough characterization of the S phase functions of SRBD1, but we hope the reviewer agrees that these experiments are beyond the scope of the current studies.

• **Statistics are missing in Fig. 6.**

We apologize for the omission. Statistics have now been added to the graphs in Figure 6, with the figure legend noting the number of replicates and the statistical tests used.

References

1. Earnshaw, W. C., Halligan, B., Cooke, C. A., Heck, M. M. & Liu, L. F. Topoisomerase II is a structural component of mitotic chromosome scaffolds. *J Cell Biol* **100**, 1706–1715 (1985).
2. Saitoh, N., Goldberg, I. G., Wood, E. R. & Earnshaw, W. C. ScII: an abundant chromosome scaffold protein is a member of a family of putative ATPases with an unusual predicted tertiary structure. *J Cell Biol* **127**, 303–318 (1994).
3. Dekker, B. & Dekker, J. Regulation of the mitotic chromosome folding machines. *Biochem J* **479**, 2153–2173 (2022).
4. Baxter, J. *et al.* Positive supercoiling of mitotic DNA drives decatenation by topoisomerase II in eukaryotes. *Science* **331**, 1328–1332 (2011).
5. Charbin, A., Bouchoux, C. & Uhlmann, F. Condensin aids sister chromatid decatenation by topoisomerase II. *Nucleic Acids Res* **42**, 340–348 (2014).

6. Brahmachari, S. & Marko, J. F. Chromosome disentanglement driven via optimal compaction of loop-extruded brush structures. *Proc Natl Acad Sci U S A* **116**, 24956–24965 (2019).
7. Finardi, A., Massari, L. F. & Visintin, R. Anaphase Bridges: Not All Natural Fibers Are Healthy. *Genes (Basel)* **11**, 902 (2020).
8. Hudson, D. F., Vagnarelli, P., Gassmann, R. & Earnshaw, W. C. Condensin is required for nonhistone protein assembly and structural integrity of vertebrate mitotic chromosomes. *Dev Cell* **5**, 323–336 (2003).
9. Ono, T. *et al.* Differential contributions of condensin I and condensin II to mitotic chromosome architecture in vertebrate cells. *Cell* **115**, 109–121 (2003).
10. Vagnarelli, P. *et al.* Condensin and Repo-Man-PP1 co-operate in the regulation of chromosome architecture during mitosis. *Nat Cell Biol* **8**, 1133–1142 (2006).
11. Samoshkin, A. *et al.* Human condensin function is essential for centromeric chromatin assembly and proper sister kinetochore orientation. *PLoS One* **4**, e6831 (2009).
12. Samejima, K. *et al.* Functional analysis after rapid degradation of condensins and 3D-EM reveals chromatin volume is uncoupled from chromosome architecture in mitosis. *J Cell Sci* **131**, jcs210187 (2018).

Response to reviewers

We thank the reviewers for their thoughtful comments. Their input has greatly improved the manuscript.

Reviewer #1 (Remarks to the Author):

The authors have addressed most of my concerns, but I see some issues still persist. A general comment is that: The authors describe the new data in support of the new mechanical insights obtained in the rebuttal, but it would have been better if they provided a lay paragraph description on what precisely the new mechanical insights were and why the findings were important.

Accurately segregating chromosomes to two daughter cells during mitosis is essential for life. To do this, cells must reorganize their chromosomes and remove any entanglements between sister chromatids. Condensin complexes are the primary drivers of mitotic chromosome organization, and critical for directing the activity of TOP2A to resolve the topological links. In this manuscript, we identified SRBD1, a largely uncharacterized protein, as a novel component of the mitotic scaffold that is essential for chromosome segregation. SRBD1 acts during early mitosis to shape mitotic chromosomes and promote their disentanglement by TOP2A. It does this by promoting proper mitotic TOP2A localization. While SRBD1 and condensin complexes are each required for TOP2A localization, disruption of both significantly improves TOP2A recruitment and activity. This synthetic rescue suggests an unexpected and intimate link between SRBD1 and condensin functions to properly organize mitotic chromosomes so they can be accurately distributed into daughter cells. Thus, our work has identified SRBD1 as essential for maintaining genome integrity during cell division and opens a new area of research into the regulation of condensin and TOP2A activity during mitosis. These findings will be of general interest to individuals studying chromosome organization, topoisomerase functions, and mitosis. In addition to these major findings, we also presented data that SRBD1 has important functions in interphase cells, specifically during DNA replication. We include those data since they were what led us to study SRBD1 and provided the first links to TOP2A through a tight correlation in SRBD1 and TOP2A abundance on nascent DNA during replication. Thus, our work will also be of interest to individuals studying DNA replication where DNA entanglements also must be managed by TOP2A. Finally, our observation that SRBD1 has structural and sequence similarity to histone chaperones as well as the ability to bind both DNA and histones provides the first indication of how this protein may act at the molecular level. As with any discovery of a previously uncharacterized protein, our results raise many important questions which will require years of work by many laboratories to answer. We modified the discussion text to more clearly articulate these points.

Other specific points are:

1. I am sorry, I still have a major issue with the live-cell data shown to claim that mitotic spindles are normal overall. Maybe the issue is that they do not have a control movie to show here. It is possible that the spindles look more or less normal until metaphase, but considering the drastic chromosomal phenotype and based on what data is shown in the live imaging experiments, what I see is that there is a major spindle phenotype later in mitosis post-metaphase. Maybe I am focusing on the overall general microtubule structure. Maybe the

authors can use specific language in the text to temper down the claims? Maybe show still pictures of anaphase/telophase spindles and/or exclude the live-imaging data? This reviewer has no difficulty in believing that spindle in later stages of mitosis could be abnormal if there is a major issue with chromosome compaction or in the case of major mis-segregation events/entanglements and bridges.

We thank the reviewer for this comment that provides us an opportunity to better clarify our result and conclusions about the mitotic spindle. As the reviewer points out, the SRBD1-deficient cells do not exhibit spindle abnormalities (mono/multipolar spindles, abnormal spindle length, unfocused spindle poles) through metaphase. We also observe normal chromosome congression and there is no apparent defect in mitotic checkpoint activity that monitors spindle attachments to the chromosomes. Thus, we do not have any evidence that SRBD1 is required for spindle assembly or function prior to anaphase. After this point, most SRBD1-deficient cells show the anaphase failure phenotype with neither anaphase A nor anaphase B occurring properly. In these cells, the spindle poles are pulled together during the failed anaphase instead of the chromosomes being pulled to the poles and poles moving away from each other (Figure S6d and Supplementary Video 6). We agree with the reviewer that this anaphase spindle behavior is abnormal, and we include a clear description of this point in the revised manuscript (see page 10, lines 211-217 and new paragraph in discussion starting on line 331). To better document the range of chromosome and spindle behaviors in SRBD1-deficient cells, we also added still images from two additional SRBD1-degron cells with less severe anaphase phenotypes (new Fig. S6e). These cells show visible chromosome segregation, but with extensive anaphase chromatin bridges. The spindle poles do move away from each other as expected in these cells. Still images and a movie from live imaging of a control cell are also provided for comparison, as recommended by the reviewer (new Fig. S6c and new Supplementary Video 5).

An important question is whether the anaphase failure in SRBD1 deficient cells is because of a defect in spindle function or whether it is a consequence of chromosome entanglements and bridges. Multiple lines of evidence suggest that there is a failure to fully decatenate the chromosomes leading to extensive entanglements and bridges. We favor that this defect is primarily responsible for the failed chromosome segregation for the following reasons:

1. Some SRBD1-deficient cells partially separate their chromosomes but contain clearly visible chromatin bridges. The mitotic spindle in these cells looks normal.
2. There is a visible attempt at chromosome segregation in most SRBD1-deficient cells that undergo anaphase failure, with a pulling force that briefly produces a modest degree of chromosome separation.
3. The mislocalization of mitotic TOP2A in SRBD1-deficient cells is consistent with the idea that persistence of DNA catenanes is a major problem.
4. The reduction in anaphase failure by condensin II depletion in the SRBD1-deficient cells is accompanied by a rescue in TOP2A localization to mitotic chromosomes, suggesting improved decatenation facilitates anaphase progression with normal anaphase spindles.
5. Inducing a prometaphase delay with nocodazole largely rescues the anaphase failure in SRBD1-deficient cells. Additional time in prometaphase might readily allow a reduced amount of TOP2A on the chromosomes to decatenate and resolve the entanglements, while it is unclear how this would rescue an anaphase spindle function defect.
6. Progression through mitosis is efficiently blocked by nocodazole, indicating that the spindle assembly checkpoint is functional in SRBD1-deficient cells.

7. SRBD1 localizes to the chromosome axis and binds histones and DNA. We do not observe localization to the spindle.

Nonetheless, we do not exclude the possibility that in addition to being required for TOP2A-dependent decatenation, SRBD1 also has another function in controlling spindle dynamics which contributes to the failure in chromosome segregation. This is explicitly stated in the revised discussion (page 14, lines 331-358).

2. I think one of the other reviewers also alluded to this in the initial review; but what I wanted with regard to the data in the new Figure 3B and S2C is for them to simply show immunofluorescence pictures of the data used for this quantification and not just the quantified data. I do not see why this should be a problem.

We added representative images of γ H2AX immunostaining as new Fig. S2d, as requested.

3. I still do not see textual details about the significance of using PICH and UBF included in the manuscript. I thought this would benefit the general readers. I was not just asking about details of the reagents used in this regard.

To provide additional information for a general reader, we expanded the explanation of these data in the main text of the paper as requested (page 7).

The significance of using PICH and UBF is also outlined below:

Ultrafine anaphase bridges cannot be visualized using DNA dyes and are instead identified by the localization of specific proteins. One such marker is PICH, a protein that binds duplex DNA under tension. This helicase recruits additional proteins to ultrafine bridges and promotes nucleosome remodeling and DNA disentanglement to aid in the resolution of ultrafine bridges and the maintenance of genome stability.

Anaphase bridges are classified by their molecular structure and often arise from specific loci. Understanding the types of bridges formed and the proteins localized to them can provide insight about the origin of these structures and the cellular processes involved in their formation and resolution. Centromeres and ribosomal DNA are two loci prone to forming anaphase bridges, therefore we identified these regions by immunostaining for CENPB and UBF, respectively. Neither locus contributed significantly to the anaphase chromatin bridges observed after SRBD1 inactivation, indicating the problems are not specific to these loci.

4. Minor: In new Figure 7E, since the authors have already quantified this data, it might be better to increase the overall brightness of the figure as this look quite dim (even in the cases where one is supposed to see bright TOP2A).

At the request of the reviewer, we altered the brightness of TOP2A in all panels in Fig. 7e to make the staining differences more apparent.

5. Minor: I did not notice this in the previous round of submission, but I am curious to know if the bottom most band in the top blot of Figure 3A which runs very close to SRDB1 in the case of mAC-SRBD1 (c1) is leaky endogenous expression or if this is possibly some other protein?

The degron cell lines used in these studies originated from single cell clones and were genotyped by PCR to confirm that both SRBD1 alleles had been successfully edited. The endogenous protein expression in these cells solely contains a degron tag. There are numerous cross-reacting proteins on SRBD1 immunoblots of total cell extracts, such as the band the reviewer is referring to in Fig. 3a. As shown in Reviewer Fig. 1 below, the band in question can clearly be observed to migrate below the band representing the untagged SRBD1 protein in the parental cell clone. The extent of separation from untagged SRBD1 varies from blot to blot, depending on run time and acrylamide percentage. We added an asterisk to the blots in Fig. 3a, Fig. S2a, and Fig. S8c to identify this band as a cross-reacting protein.

Reviewer Figure 1. The SRBD1 antibody cross-reacts with other proteins in whole-cell extracts. Immunoblot analysis of SRBD1 (parental clone) and mAID2-mClover-SRBD1 (mAC-SRBD1, clone 1) in asynchronous cells treated with DMSO or 1 μ M 5-Ph-IAA for the indicated amounts of time. The untagged SRBD1 protein is expressed only in the parental cell clone and is unaffected by the addition of 5-Ph-IAA. mAC-SRBD1 clone 1 solely expresses a degron-tagged SRBD1 protein, as indicated by the expected change in molecular weight and rapid degradation after the addition of 5-Ph-IAA. A cross-reacting protein that migrates just below the untagged SRBD1 protein is denoted by an asterisk (*).

Reviewer #2 (Remarks to the Author):

The revised manuscript provided some additional data to further define the mechanisms and/or functions of SRBD1 in mitosis. The key data is that although SRBD1 and CAP-D3 depletion each reduced TOP2A chromatin association, double depletion of these proteins somehow increased TOP2A chromatin loading, which the authors hypothesized to be the mechanism underlying the mitotic phenotypes observed in cells with SRBD1 depletion.

These data seem to disagree with their previous observation that depletion of TOP2A was also able to partially rescue the mitotic phenotypes observed in cells with SRBD1 depletion. They have now removed these data in the revised manuscript.

The concern is that CAP-D3 depletion by siRNAs took at least two days. It remains possible that the rescue phenotypes observed in these double depletion experiments may be due to some indirect effects on cell proliferation and/or mitotic progression. Since the authors would like to use this unique genetic interaction to draw their main conclusion, i.e. SRBD1 facilitates chromosome segregation by promoting TOP2A localization to mitotic chromosomes, I agree with the other reviewer that inducible co-depletion of these two proteins may be needed to further verify this interesting genetic interaction.

To address whether the rescue effect was due to an indirect effect on cell proliferation or mitotic progression, we examined cell cycle profiles of control and condensin-depleted cells 48 hours after siRNA transfection. The CAP-D3 knockdown cells are almost identical to the control (new Fig. S8b). CAP-D3 depletion does produce a slight increase in mitotic cells (3.5% compared to 2.7%). This increase may represent a minor delay in mitotic progression, as would be expected for condensin inactivation. However, this potential delay is not likely contributing to the rescue phenotypes since the cell cycle profiles following depletion of CAP-H and CAP-D3 are similar, and yet the rescue is specific to condensin II depletion. Furthermore, the extent of the increase in mitotic cells did not correlate with the extent of mitotic improvement, since depletion of CAP-H2 produced a greater increase in mitotic cells while CAP-D3 depletion produced better mitotic outcomes. These results suggest that the improvement in anaphase progression and mitotic TOP2A localization observed following SRBD1 inactivation in condensin II-depleted cells is not likely an indirect effect of cell proliferation or mitotic progression. We also note that our main conclusion that SRBD1 is needed to properly localize TOP2A to facilitate chromosome segregation is not dependent on the results from the genetic interaction between condensin II and SRBD1. Certainly, we agree that it would be nice to have a double-degron cell line. As we explained to reviewer 3 in the previous response letter, we have not yet been able to generate that reagent.

Reviewer #3 (Remarks to the Author):

The authors have made thorough revisions and addressed all my comments comprehensively. I believe that the revised version of the manuscript has significantly improved. The revisions have strengthened the clarity and rigor of the work, and I am satisfied with the changes made. I have no further comments or suggestions at this time.

In my opinion, the manuscript is now suitable for publication in Nature Communications.

We thank the reviewer for their insightful comments and suggestions, which have significantly improved this manuscript.

Response to reviewers

Reviewer #1 (Remarks to the Author):

I am satisfied with the revisions. It would be good to include relevant details from the authors response to my 1st query (lay paragraph description of significance) in the discussion section of the manuscript if they have not already done so.

We thank the reviewer again for helping us to clarify key points in the manuscript. As suggested, we have added relevant details from the lay paragraph description to the discussion section of the revised manuscript.

Reviewer #2 (Remarks to the Author):

The revised manuscript addressed some of my concerns. Unfortunately the authors do not have inducible co-depletion cells to further support their working hypothesis. Nevertheless, the authors performed siRNA mediated depletion of CAP-H2 and CAP-D3 in their SRBD1 inducible cell line. They should be able to comment on whether CAP-H2 or CAH-D3 depletion is able to rescue cell lethality caused by SRBD1 depletion.

We thank the reviewer again for their helpful comments. While condensin II depletion can reduce the severity of the anaphase defects in the first mitosis after SRBD1 degradation, loss of condensin cannot rescue the viability defect resulting from SRBD1 inactivation (Reviewer Fig. 1). We added this data to the manuscript as Supplementary Fig. 8c, d.

Reviewer Figure 1. Condensin II depletion does not rescue the viability defect of SRBD1-deficient cells. SRBD1 degron cells were transfected with siRNAs targeting condensin I (CAP-H) or condensin II (CAP-H2 and CAP-D3). Non-targeting (NT) siRNA was used as a control. Viability was measured by colony forming ability after a single treatment of DMSO or 1 μ M 5-Ph-IAA at 48 hr after siRNA transfection. Representative colony images (a) and quantitation (b) are shown. Graph displays the mean \pm SD of three technical replicates.